# Fairness-Aware Dense Subgraph Discovery

**Emmanouil Kariotakis**                                    *emmanouil.kariotakis@kuleuven.be*
*ESAT-STADIUS*
*KU Leuven*
**Nicholas D. Sidiropoulos**                                    *nikos@virginia.edu*
*Department of Electrical Engineering*
*University of Virginia*
**Aritra Konar**                                    *aritra.konar@kuleuven.be*
*ESAT-STADIUS*
*KU Leuven*

**Reviewed on OpenReview:** *https://openreview.net/forum?id=7rqV7Cb67L*

## Abstract

Dense subgraph discovery (DSD) is a key graph mining primitive with myriad applications including finding densely connected communities which are diverse in their vertex composition. In such a context, it is desirable to extract a dense subgraph that provides fair representation of the diverse subgroups that constitute the vertex set while incurring a small loss in terms of subgraph density. Existing methods for promoting fairness in DSD have important limitations - the associated formulations are NP–hard in the worst case and they do not provide flexible notions of fairness, making it non-trivial to analyze the inherent trade-off between density and fairness. In this paper, we introduce two tractable formulations for fair DSD, each offering a different notion of fairness. Our methods provide a structured and flexible approach to incorporate fairness, accommodating varying fairness levels. We introduce the fairness-induced *relative* loss in subgraph density as a *price of fairness* measure to quantify the associated trade-off. We are the first to study such a notion in the context of detecting fair dense subgraphs. Extensive experiments on real-world datasets demonstrate that our methods not only match but frequently outperform existing solutions, sometimes incurring even less than half the subgraph density loss compared to prior art, while achieving the target fairness levels. Importantly, they excel in scenarios that previous methods fail to adequately handle, i.e., those with extreme subgroup imbalances, highlighting their effectiveness in extracting fair and dense solutions.

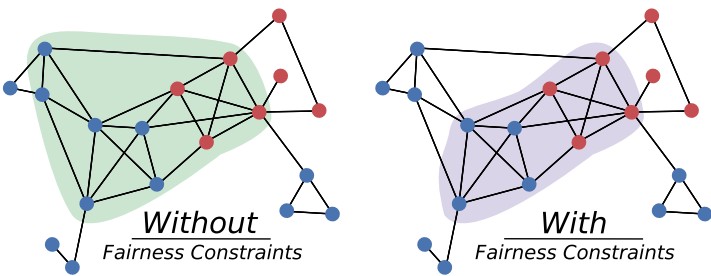

Figure 1: A toy example of Fair DSG with 2 vertex groups; protected (red) and unprotected (blue). Left: Densest subgraph *without* fairness constraints; density = 2. Right: Densest subgraph *with* fairness constraints (equal number of red and blue vertices); density = 1.875.

# 1   Introduction

Ensuring that data-driven algorithms do not disproportionately impact different population subgroups is a key challenge in human-centered applications of artificial intelligence. The field of *algorithmic fairness* (Dwork et al., 2012; Feldman et al., 2015; Kleinberg et al., 2018; Holstein et al., 2019) focuses on designing fairness-enhancing mechanisms to mitigate algorithmic bias. Fairness considerations have been formulated as optimization problems in both supervised and unsupervised learning (see the surveys Friedler et al. (2019); Chouldechova & Roth (2020) and references therein). In contrast, it remains a nascent area of research in graph mining, where fairness considerations are also important (Dong et al., 2023). To date, only few graph mining problems have been analyzed under the lens of algorithmic fairness.

Dense subgraph discovery (DSD) is one key graph mining primitive that aims to extract subgraphs with high internal connectivity from a given graph (see the survey Lanciano et al. (2024) and references therein). DSD finds widespread applications ranging from mining social media trends (Angel et al., 2012), detecting patterns in gene annotation graphs (Khuller & Saha, 2009), and spotting fraud in e-commerce and financial networks (Hooi et al., 2016; Li et al., 2020; Chen & Tsourakakis, 2022). A popular formulation used for extracting dense subgraphs maximizes the average induced degree (Goldberg, 1984). This is because the problem can be solved exactly using maximum-flows, or approximated efficiently at scale using greedy vertex-peeling algorithms (Charikar, 2000; Boob et al., 2020; Chekuri et al., 2022), or convex optimization algorithms (Harb et al., 2023; Nguyen & Ene, 2024; Harb et al., 2022), with guarantees on the sub-optimality of the solution.

In this paper, we study the problem of extracting the densest subgraph from a given graph that meets a target fairness criterion. Such considerations can arise in DSD when the vertices of the graph (e.g., corresponding to a web or social network) are annotated with sensitive information pertaining to an individual's gender/race/religion/political leaning, etc., which partitions the vertex set into different population subgroups. The fair dense subgraph problem then corresponds to locating a dense subgraph with adequate levels of representation of each subgroup in the extracted subset. Simply applying a pre-existing algorithm for DSD can fail in this regard. Indeed, studies on real-world graphs (Anagnostopoulos et al., 2020; 2024) have revealed that existing methods return subgraphs which typically exhibit strong homophily, with little to no diversity in the composition of the vertex attributes. Motivating examples of fairness in DSD include selection of diverse teams from a social network of professional contacts (Marcolino et al., 2013; Rangapuram et al., 2013), and recommending balanced content to social media users that spans the various views of the individuals comprising the network, thus mitigating polarization (Musco et al., 2018).

While DSD is a well-studied topic, extracting *fair* dense subgraphs has received limited attention. Although the recent works of Anagnostopoulos et al. (2020; 2024); Miyauchi et al. (2023) have initiated progress in this area, they have some inherent limitations. Notably, their formulations are NP–hard in the worst-case, necessitating approximation algorithms. These methods also struggle with rigid or hard-to-set fairness criteria. Another key issue is the inherent trade-off between subgraph density and fairness, which is difficult to analyze and has not been formally explored due to the NP-hardness of existing approaches (see Section 2 for details).

**Contributions:** In this paper, we focus on the setting where the vertices of the graph are divided in two subgroups, protected and unprotected. Our objective is to overcome the limitations of prior work by **(i)** proposing new, tractable (polynomial-time solvable) formulations which are **(ii)** capable of generating a spectrum of fair dense subgraphs with varying levels of fairness. This approach enables users to analyze the trade-off between subgraph density and fair representation of the protected group, and quantify the *price of fairness*, i.e., the density loss required to meet a target fairness level. We point out that these features are absent in the formulations of Anagnostopoulos et al. (2020; 2024); Miyauchi et al. (2023). Our key technical idea is to promote fairness in DSD through a pair of judiciously chosen regularizers that create tractable problems and establish regularization paths representing various fairness levels.

To summarize:

- We introduce two *tractable* formulations for fair DSD that are capable of accommodating *variable fairness levels*. This enables flexible selection across a spectrum of target fairness levels, enhancing the applicability of the formulations.

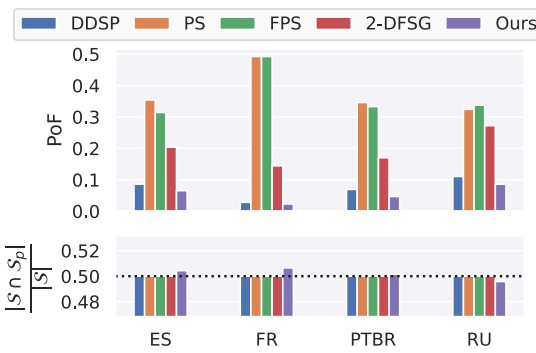

Figure 2: Comparing perfectly balanced fair subgraphs obtained via prior art and FADSG-I (ours, purple) on 4 different TWITCH datasets. (Top): Price of fairness (the lower, the better). (Bottom): Fraction of protected vertices in induced subgraph (set to 50% for perfect balance).

- We analyze the inherent trade-off between subgraph density and target fairness for a difficult example using the *price of fairness* measure. Our results indicate that enhancing fairness can significantly reduce density, regardless of the algorithm used.

- Through extensive experiments on diverse datasets, we demonstrate *superior performance* (see Figure 2) and *practical utility* of our formulations over existing approaches.

**Preliminaries and Notation:** Consider a simple, undirected graph $\mathcal{G} = (\mathcal{V}, \mathcal{E})$ on $n := |\mathcal{V}|$ vertices and $m := |\mathcal{E}|$ edges. Let $\mathbf{w} : \mathcal{E} \to \mathbb{R}^m_{++}$ denote a set of positive weights defined on the edges of $\mathcal{G}$; i.e., each edge $\{i, j\} \in \mathcal{E}$ is assigned a positive weight $w_{\{i,j\}}$. If each edge has unit weight, the graph $\mathcal{G}$ is said to be *unweighted*. Let $\mathbf{A}$ denote the $n \times n$ symmetric adjacency matrix of $\mathcal{G}$ with entries $a_{ij} = w_{\{i,j\}}, \forall \{i, j\} \in \mathcal{E}$ and $\mathbf{e}$ be the $n$-dimensional vector of all-ones. Given a vertex subset $\mathcal{S} \subseteq \mathcal{V}$, let $\mathcal{G}_{\mathcal{S}} = (\mathcal{S}, \mathcal{E}_{\mathcal{S}})$ denote the subgraph induced by $\mathcal{S}$. The (weighted) sum of edges in $\mathcal{G}_{\mathcal{S}}$ is denoted as $e(\mathcal{S}) := \sum_{\{i,j\} \in \mathcal{E}_{\mathcal{S}}} w_{\{i,j\}}$. The subgraph density of $\mathcal{G}_{\mathcal{S}}$ is its average (induced) degree, which is expressed as $\rho(\mathcal{S}) := 2e(\mathcal{S})/|\mathcal{S}|$. Given a vertex subset $\mathcal{S} \subseteq \mathcal{V}$, let $\mathbf{x} \in \{0, 1\}^n$ denote its corresponding binary indicator vector. In terms of $\mathbf{x}$, the subgraph density can be equivalently expressed as $\rho(\mathbf{x}) := (\mathbf{x}^\top \mathbf{A} \mathbf{x})/(\mathbf{e}^\top \mathbf{x})$. In addition, each vertex $v \in \mathcal{V}$ possesses a binary group label that designates it as belonging to the protected or unprotected group. Thus, the vertices of $\mathcal{G}$ can be partitioned into a protected $\mathcal{S}_p \subset \mathcal{V}$, with $n_p := |\mathcal{S}_p|$, and an unprotected subset $\bar{\mathcal{S}}_p := \mathcal{V} \setminus \mathcal{S}_p$. A real-valued set function $f : 2^{\mathcal{V}} \to \mathbb{R}$ defined on a ground set $\mathcal{V}$ is *supermodular* if and only if (iff) $f(\mathcal{A}) + f(\mathcal{B}) \le f(\mathcal{A} \cup \mathcal{B}) + f(\mathcal{A} \cap \mathcal{B})$ for all $\mathcal{A}, \mathcal{B} \subseteq \mathcal{V}$. Equivalently, $f$ is supermodular iff $f(v|\mathcal{A}) \le f(v|\mathcal{B})$, for all $\mathcal{A} \subseteq \mathcal{B} \subseteq \mathcal{V} \setminus \{v\}$, where $f(v|\mathcal{S}) := f(\mathcal{S} \cup \{v\}) - f(\mathcal{S})$ denotes the marginal value of $v$ with respect to $\mathcal{S}$. If $f(\emptyset) = 0$, $f$ is said to be normalized. A function $f$ is submodular iff $-f$ is supermodular. A function is modular if it is both supermodular and submodular. Relevant to this paper, the function $e(\mathcal{S})$ is monotone supermodular, whereas $|\mathcal{S}|$ is modular.

## 2 Related Work

**Densest subgraph and it variants:** Given a graph $\mathcal{G}$, the classic DENSEST SUBGRAPH (DSG) problem (Goldberg, 1984) aims to extract a dense vertex subset $\mathcal{S} \subseteq \mathcal{V}$ that maximizes the average (induced) degree. DSG can be solved exactly in polynomial time via maximum flows, but this is computationally demanding for large graphs. In practice, a scalable greedy algorithm that iteratively peels vertices is used, providing a $1/2$-factor approximation (Asahiro et al., 2000; Charikar, 2000). The algorithm was generalized in the recent work of Boob et al. (2020), which proposed a multi-stage greedy algorithm GREEDY++ that exhibits fast convergence to a near-optimal solution of DSG without relying on flows. The follow-up work of Chekuri et al. (2022) provided a performance analysis of GREEDY++, and demonstrated that it can attain $(1 - \epsilon)$-approximate solutions of DSG in $O(1/\epsilon^2)$ iterations. Building on this approach, a general peeling-based

framework SUPER-GREEDY++ was introduced by Chekuri et al. (2022) for obtaining $(1 - \epsilon)$-approximate solutions for the broader problem class of computing densest supermodular subsets, which can be expressed as $\max_{\mathcal{S} \subseteq \mathcal{V}}\{F(\mathcal{S})/|\mathcal{S}|\}$, where $F(\mathcal{S})$ is a monotone, normalized, supermodular function. Note that DSG is a special case, with $F(\mathcal{S}) = e(\mathcal{S})$. More generally, it was shown by Harb et al. (2023) that SUPER-GREEDY++ exhibits convergence to an optimal dense decomposition vector of $f$, which corresponds to a partition of of $\mathcal{V}$ into a finite collection of subsets $\mathcal{S}_1, \ldots, \mathcal{S}_k$, such that each for each $i \in \{1, \ldots, k\}$, $\mathcal{S}_i$ is the unique maximal subset that maximizes $\lambda_i = (F(\mathcal{S}_i \cup \mathcal{X}_{i-1}) - F(\mathcal{X}_{i-1}))/|\mathcal{S}_i|$ over $\mathcal{V} \setminus \mathcal{X}_{i-1}$, where $\mathcal{X}_{i-1} := \cup_{j<i} S_j$. The value of $\lambda_1$ corresponds to density of the densest supermodular subset, whereas subsequent values monotonically decrease, $\lambda_1 > \cdots > \lambda_k$. For DSG, we refer to such a decomposition as a dense decomposition on a graph $\mathcal{G}$.

A separate line of research (Danisch et al., 2017; Harb et al., 2022; Nguyen & Ene, 2024) for developing fast approximation algorithms for DSG has blossomed around applying convex-optimization algorithms for solving a tight linear programming (LP) relaxation of DSG originally proposed by Charikar (2000). The work of Danisch et al. (2017) considered a quadratic programming problem derived from the dual of the LP relaxation of Charikar (2000), and applied the Frank-Wolfe algorithm (Frank et al., 1956) for solving it. Recently, the authors of Harb et al. (2022) developed faster algorithms for DSG and dense decompositions on graphs by solving the dual LP relaxation via accelerated proximal gradient (Beck & Teboulle, 2009). This was later improved upon by Nguyen & Ene (2024) using tools from area convexity Sherman (2017) and accelerated random coordinate descent Ene & Nguyen (2015) to develop faster approximation algorithms for DSG and dense decompositions on graphs, respectively.

An important variant of DSG is finding the $k$-core decomposition of a graph Seidman (1983). A $k$-core is the maximal subgraph (by size) where every vertex has an induced degree of at least $k$. The largest value of $k$ for which such a subgraph exists is known as the degeneracy, and the corresponding subgraph is known as the max-core. Extracting the max-core is tantamount to finding the vertex subset which maximizes the *minimum* induced degree. A slight modification of the greedy peeling algorithm for DSG exactly solves this problem.

Recently, it has been shown (Veldt et al., 2021) that the solution of DSG and the $k$-core both correspond to special cases of a more general framework known as the GENERALIZED MEAN DENSEST SUBGRAPH (GMDSG) problem. This framework is parameterized by a single parameter $p$, based on computing generalized means of degree sequences of a subgraph, with DSG and $k$-core corresponding to the choices of $p = 1$ and $p = -\infty$, respectively. Hence, by varying $p$, one can extract a family of dense subgraphs which obey different notions of density. For $p \geq 1$, GMDSG can be solved optimally in polynomial-time via maximum flows, or approximated via a generalized greedy peeling algorithm. Additionally, it was shown by Chandrasekaran et al. (2024) that for $p < 1$, GMDSG is NP–hard for $p$ in the range $p \in (-1/8, 0) \cup (0, 1/4)$, and that the solutions of DSG and max-core provide a $1/2$-approximation for GMDSG when $p < 1$.

Other variants of DSG consider imposing size restrictions on the extracted subgraph. The DENSEST-$k$-SUBGRAPH (D$k$S) problem (Feige et al., 2001) seeks to determine the edge densest subgraph of size $k$, which is NP–hard and also notoriously difficult to approximate (Bhaskara et al., 2012; Manurangsi, 2017). Notwithstanding this fact, practical algorithms which work well for this problem on real graphs include Papailiopoulos et al. (2014); Konar & Sidiropoulos (2021). Another variant, Dam$k$S (Andersen & Chellapilla, 2009), maximizes average degree subject to the size of the subgraph being *at most* $k$, and is NP–hard. Meanwhile, the Dal$k$S problem (Andersen & Chellapilla, 2009) maximizes average degree subject to the subgraph size being *at least* $k$. Dal$k$S is also NP–hard (Khuller & Saha, 2009), with greedy and linear programming approximation algorithms known.

**Fairness in DSD:** Subgraphs extracted via DSG or its variants are not guaranteed to fulfill a pre-specified fairness criterion (regardless of the algorithmic approach employed), as the respective formulations do not explicitly take into account any protected attributes that the vertices may possess. This issue was recently brought up by Anagnostopoulos et al. (2020; 2024); Miyauchi et al. (2023), which considered the task of extracting dense subgraphs that satisfy a group-fairness criterion. In Anagnostopoulos et al. (2020; 2024), each vertex is a member of either a protected or an unprotected group, and the goal is to extract the densest subgraph with an equal number of vertices from both groups (*perfect balance*). The follow-up work of Miyauchi et al. (2023) proposed a pair of fairness promoting formulations which can handle multiple vertex groups. The first (Miyauchi et al., 2023, Problem 1) corresponds to an extension of Anagnostopoulos et al. (2020;

2024) that aims to find the densest subgraph subject to the constraint that the number of vertices from each group in the solution does not exceed a fixed fraction of the solution size. The second formulation (Miyauchi et al., 2023, Problem 2) concerns finding the densest subgraph while ensuring that at least a pre-specified number of vertices from each group are present in the solution. A separate work (Oettershagen et al., 2024) considered a variation of the fair DSG problem where the edge relationships (as opposed to vertices) in a graph are modeled as belonging to different groups. The authors of Oettershagen et al. (2024) proposed formulations for extracting fair dense subgraphs containing *exactly*, *at most*, and *at least* a certain number of edges from each edge type.

While serving as a reasonable means of promoting fairness in DSD, the aforementioned approaches have important limitations. **(i):** Fairness is enforced using hard combinatorial constraints, which renders the problem formulations of Anagnostopoulos et al. (2020; 2024); Miyauchi et al. (2023) NP–hard in the worst case. Meanwhile, the decision versions of the problems considered in Oettershagen et al. (2024) are demonstrated to be NP-complete. Since optimal solution cannot be guaranteed in polynomial-time for every problem instance, it is a non-trivial task to analyze the trade-off between density and fairness made by these formulations on real-world datasets. **(ii):** The approximation algorithms developed in Anagnostopoulos et al. (2020; 2024); Miyauchi et al. (2023) do not tackle the problem directly, but adopt a two-pronged approach. In the first part, which corresponds to a relaxation step, the fairness constraints are decoupled from the problem formulation and a dense subgraph is extracted. This is followed by a "rounding" step that takes the dense subgraph as input and aims to make the solution feasible with respect to the fairness constraints. However, such a strategy, which is employed in Anagnostopoulos et al. (2020; 2024) and Miyauchi et al. (2023, Problem 1), can cause a large drop in practical performance. Although the spectral relaxation algorithms proposed in Anagnostopoulos et al. (2020; 2024) provide sub-optimality guarantees, these apply under the assumption that the degree distribution of the underlying graph is near uniform; a condition that rarely holds in real-world graphs which typically exhibit a skewed-degree distribution (Newman, 2003). In practice a heuristic procedure adjusts the output of DSG by ad-hoc addition of vertices from the least represented group until perfect balance is reached. Meanwhile, Miyauchi et al. (2023) compute an approximation to Dal$k$S to obtain an initial solution for Problem 1 without imposing fairness, which is then refined via a post-processing step that adds or removes vertices until the target fairness constraints are met. Regarding the second formulation of Miyauchi et al. (2023, Problem 2), a linear programming relaxation with quality guarantees is developed, albeit the algorithm exhibits high complexity. A low complexity greedy algorithm is proposed as an alternative, which offers quality guarantees under very specific conditions on the optimal solution of the problem, which are difficult to verify *a priori*. On the other hand, Oettershagen et al. (2024) provides a constant-factor approximation algorithm for their edge-fairness problem, but these guarantees apply to graphs which are everywhere sparse – a restrictive assumption. **(iii):** The fairness criteria employed in Anagnostopoulos et al. (2020; 2024); Miyauchi et al. (2023) can be restrictive, or difficult to properly tune in order to achieve a target representation level. The case of perfect balance is the sole consideration of Anagnostopoulos et al. (2020; 2024), whereas Miyauchi et al. (2023, Problem 1) only guarantee that the representation level of each subgroup is *no more* than a fixed fraction of the total vertices in the solution. However, it is not possible to specify *a priori* which subgroup attains maximum representation or whether this upper bound is attained at all. This implies that even in the simplest setting of two subgroups (i.e., protected and unprotected), the method may not be effective in guaranteeing a target level of representation of the protected group. The second formulation Miyauchi et al. (2023, Problem 2) ensures only that *at least* a certain number of vertices from the protected group are included in the solution, and cannot guarantee attaining a specific proportion of protected vertices, which is the focus of our work. Meanwhile, Oettershagen et al. (2024) quantifies fairness w.r.t. edges via edge-type constraints, which is not directly comparable with our work as we measure fairness w.r.t. vertices.

## 3 Problem Statement

Recall the classic DENSEST SUBGRAPH (DSG) problem (Goldberg, 1984). Given an undirected graph $\mathcal{G}$, DSG aims to extract a dense vertex subset $\mathcal{S} \subseteq \mathcal{V}$ that maximizes the average (induced) degree. The problem can be expressed as

**Definition 1** (DSG).

$$\mathbf{x}^* = \underset{\mathbf{x} \in \{0,1\}^n}{\operatorname{argmax}} \ \rho(\mathbf{x}), \tag{1}$$

where $\rho(\mathbf{x}) := (\mathbf{x}^\top \mathbf{A} \mathbf{x})/(\mathbf{e}^\top \mathbf{x})$ denotes the subgraph density of a candidate subgraph $\mathbf{x} \in \{0,1\}^n$ in terms of average induced-degree.

In contrast to prior work on fairness in DSD where fairness requirements are explicitly enforced as constraints, we adopt a regularization based approach, where group-fairness in the extracted subgraph is promoted by an appropriately chosen regularizer. In this section, we introduce our formulations.

Given a graph $\mathcal{G}$ with designated protected and unprotected subgroups of vertices, let $\mathbf{e}_p \in \{0,1\}^n$ denote the binary indicator vector of the protected subset $\mathcal{S}_p$. Define the regularizer $r_1(\mathbf{x}) := (\mathbf{e}_p^\top \mathbf{x})/\mathbf{e}^\top \mathbf{x}$ and consider the first FAIRNESS-AWARE DENSEST SUBGRAPH (FADSG-I) problem.

**Definition 2** (FADSG-I).

$$\mathbf{x}^*(\lambda) = \underset{\mathbf{x} \in \{0,1\}^n}{\operatorname{argmax}} \left\{ g(\mathbf{x}, \lambda) := \rho(\mathbf{x}) + \lambda \cdot r_1(\mathbf{x}) \right\}. \tag{2}$$

Here, $\lambda \geq 0$ is the regularization parameter. The objective function $g(\mathbf{x}, \lambda)$ consists of two terms - the first term $\rho(\mathbf{x})$ corresponds to subgraph density, whereas for $\lambda > 0$, $r_1(\mathbf{x})$ promotes the inclusion of vertices from the protected subset $\mathcal{S}_p$. This can be seen by equivalently expressing $r_1(\cdot)$ as the set function

$$r_1(\mathcal{S}) = \frac{|\mathcal{S} \cap \mathcal{S}_p|}{|\mathcal{S}|}. \tag{3}$$

It is evident that $0 \leq r_1(\mathcal{S}) \leq 1$, with the upper bound being attained for any subset $\mathcal{S} \subseteq \mathcal{S}_p$. Meanwhile, $r_1(\mathcal{S}) = 0$ iff $\mathcal{S}$ does not contain any protected vertices. Thus, larger values of $r_1(\cdot)$ correspond to vertex subsets containing a greater fraction of protected vertices. Note that $r_1(\mathcal{S}) = 1/2$ corresponds to the case of *perfect balance* (Anagnostopoulos et al., 2020; 2024), with equal number of protected and unprotected vertices in $\mathcal{S}$. Moreover, the choice of $r_1(\mathcal{S}) = n_p/n$ implies that the number of vertices in the solution is proportional to the number of vertices in the population, a property known as *demographic parity* (Dwork et al., 2012). By varying $\lambda$, we allow $r_1(\cdot)$ to span a range of values, which affords flexibility in extracting dense subgraphs exhibiting varying fairness levels.

When $\lambda = 0$, FADSG-I reduces to DSG, the solution of which is not guaranteed to contain any protected vertices. As $\lambda$ is increased, greater emphasis is placed on maximizing $r_1(\mathbf{x})$, enhancing the inclusion of protected vertices. For $\lambda \to \infty$, $r_1(\mathbf{x}^*(\lambda)) \to 1$, and the solution of problem (2) corresponds to the *densest subset* of $\mathcal{S}_p$. Clearly, by varying $\lambda$ in the interval $[0, \infty)$, the solution of (2) spans a spectrum which lies between these two extremes. By appropriately selecting $\lambda$, desired representation levels of the protected subset can be achieved. Hence, in contrast to Anagnostopoulos et al. (2020; 2024); Miyauchi et al. (2023), our group-fairness formulation (2) affords much greater flexibility in trading-off subgraph density in exchange for protected subset representation.

It is worth pointing out that in certain scenarios, the solution of FADSG-I may not provide satisfactory levels of representation of the protected subset. Although the regularizer $r_1(\cdot)$ promotes the inclusion of protected vertices in the solution of (2), it does not explicitly take into account the fraction of such vertices that are represented. Even for (infinitely) large values of $\lambda$ in (2), the maximal level of representation attainable corresponds to the densest subset of $\mathcal{S}_p$. However, this does not imply that the fraction of the vertices of $\mathcal{S}_p$ that will be included in the solution of FADSG-I will be large (please see Table 7, Appendix H.3 for experimental results indicating this). These considerations motivate a more focused formulation for incorporating protected vertices in the solution, which we designate as the FADSG-II problem.

**Definition 3** (FADSG-II).

$$\mathbf{x}^*(\lambda) \in \underset{\mathbf{x} \in \{0,1\}^n}{\operatorname{argmax}} \left\{ h(\mathbf{x}, \lambda) := \rho(\mathbf{x}) - \lambda \cdot r_2(\mathbf{x}) \right\}. \tag{4}$$

In this case, $r_2(\mathbf{x}) := (\|\mathbf{x} - \mathbf{e}_p\|_2^2)/\mathbf{e}^\top \mathbf{x}$ is a regularizer and $\lambda \geq 0$ is a regularization parameter. The above problem differs from FADSG-I in the choice of the regularizer. Note that the numerator of $r_2(\cdot)$ can be viewed as the Hamming distance between the sets $\mathcal{S}$ and $\mathcal{S}_p$, since $\|\mathbf{x} - \mathbf{e}_p\|_2^2 = \sum_{i \in \mathcal{V}}(\mathbf{x}(i) - \mathbf{e}_p(i))^2$ corresponds to the number of disagreements between $\mathcal{S}$ (with indicator vector $\mathbf{x}$) and $\mathcal{S}_p$ (with indicator vector $\mathbf{e}_p$). Hence, for a given value of $\lambda > 0$, the objective function of (4) aims to maximize subgraph density while penalizing the distance between the extracted and the protected subgraphs. As $\lambda \to \infty$, $r_2(\mathbf{x}^*(\lambda)) \to 0$, and the solution $\mathbf{x}^*(\lambda)$ coincides with the *entire protected subset itself*. Building on this observation, consider the case where the target fairness requirement stipulates that 50% of the vertices in the protected subset be included in $\mathcal{S}$. We have the following result (see Appendix A.1 for proof).

**Lemma 4.** $r_2(\mathcal{S}) = 1 \Leftrightarrow \frac{|\mathcal{S} \cap \mathcal{S}_p|}{|\mathcal{S}_p|} = \frac{1}{2}$.

More generally, although problems (1) and (3) promote different notions of fairness, it is interesting to note that the two formulations are not completely unrelated. In particular, we have shown that the following relationship holds (see Appendix A.2 for the proof).

**Lemma 5.** *For the regularizers $r_1(\mathcal{S})$ and $r_2(\mathcal{S})$, the following inequalities hold: $r_1(\mathcal{S}) + r_2(\mathcal{S}) \geq 1$ and $2r_1(\mathcal{S}) + r_2(\mathcal{S}) \leq 1 + \frac{n_p}{2}$.*

The implication of the above result is that given the value of one regularizer, we can attain bounds on the other (see Figure 9, Appendix A.2 for a visualization). Since providing explicit representation of the protected subset is absent in the fairness formulations of Anagnostopoulos et al. (2020; 2024); Miyauchi et al. (2023), FADSG-II goes beyond what is attainable using the prior art.

**Tractability:** A limitation of the formulations of Anagnostopoulos et al. (2020; 2024); Miyauchi et al. (2023) is that they are NP–hard in the worst-case. Since optimality cannot be guaranteed, this implies that the solution obtained by their approximation algorithms may not be satisfactory in terms of the loss in density incurred for a target fairness level. In contrast, it turns out that problems (2) and (4) can always be solved exactly in polynomial-time, irrespective of $\lambda$ (see Appendix B for details). This implies that by varying $\lambda$, we can trace the Pareto-optimal boundary between the two components comprising the objective function (i.e., density and fairness) of both problems (2) and (4). This enables us to reveal the underlying trade-off between density and fairness on real-world datasets. As mentioned before, the intrinsic hardness of the formulations in Anagnostopoulos et al. (2020; 2024); Miyauchi et al. (2023) does not allow for such a feature.

Our formulations are explicitly designed to study the density-fairness trade-off in a disciplined way for the foundational setting of two vertex groups. As we will see later, they can outperform existing two-group (protected, unprotected) approaches by a significant margin. The extension of our approaches to the case of multiple vertex groups is deferred to future work.

## 4 Selecting the Regularization Parameter $\lambda$

A key component of problems (2) and (4) is the choice of the $\lambda$ parameter. Clearly, increasing the value of $\lambda$ promotes the inclusion of protected vertices in the subgraph extracted by FADSG-I and II, albeit at the expense of subgraph density. This section examines how $\lambda$ can be selected in our formulations to guarantee that the solutions obtained correspond to the densest subgraph that attains a target fairness level.

The first step is to determine the start and end points of the regularization paths of problems (2) and (4), which determine the intervals characterizing all possible solutions generated by each formulation. While $\lambda = 0$ corresponds to the start of the regularization path for both problems (note that this is tantamount to solving DSG in both cases), determining the end is a non-trivial task in general. This stems from the fact that both problems are combinatorial in nature. In the context of FADSG-I, we wish to determine $\lambda_{\max}$ such that for all $\lambda \geq \lambda_{\max}$, the solution $\mathbf{x}^*(\lambda)$ is the densest subset of the protected subset $\mathcal{S}_p$. Let $\mathcal{S}_p^* \subseteq \mathcal{S}_p$ denote the densest protected subset and $\rho(\mathcal{S}_p^*)$ denote its density. While it is difficult to pin down the exact analytical form of $\lambda_{\max}$, the following result can be derived (see Appendix C.1 for proof).

**Proposition 6.** *For $\lambda \geq \lambda_{\max} := [\max_{\mathcal{S} \supset \mathcal{S}_p}\{(\rho(\mathcal{S}) - \rho(\mathcal{S}_p^*))/(1 - \frac{n_p}{|\mathcal{S}|})\}]$, the solution of (2) is the densest protected subset $\mathcal{S}_p^*$.*

The above result shows that there exists a finite value of $\lambda$ that corresponds to the end of the regularization path of FADSG-I. However, it is difficult to further simplify it. Meanwhile, for FADSG-II, the analogous definition of $\lambda_{\max}$ corresponds to the value such that for all $\lambda \geq \lambda_{\max}$, the solution $\mathbf{x}^*(\lambda)$ is the protected subset $\mathcal{S}_p$. For the end of its regularization path, a finite $\lambda$ can be shown to exist, similarly to FADSG-I (see Appendix C.2). In practice, it is computationally easier to find a suitable $\lambda_{\max}$ (which we know exists owing to Propositions 6 and 10 in Appendix C) via trial and error (see Appendix G, Table 6 for values used). This enables us to bound the interval of the regularization path for a given dataset, which in turn facilitates the selection of the regularization parameter that corresponds to a target fairness level, as explained next.

### 4.1 Selection of Target Fairness Level

**FADSG-I:** For $\lambda = 0$, let $\mathbf{x}^*(0)$ denote the solution of problem (2), i.e., DSG. We also denote $l_1 := r_1(\mathbf{x}^*(0)) \geq 0$ as the fraction of protected vertices present in the solution of DSG. This corresponds to the level of fairness (as measured by $r_1(\cdot)$) that is attainable by the solution of the non-fair DSG problem, and is not guaranteed to be non-zero in general. Meanwhile, we know that $\mathbf{x}^*(\lambda_{\max})$ corresponds to the densest protected subset for which $u_1 := r_1(\mathbf{x}^*(\lambda_{\max})) = 1$. Let $\alpha \in [l_1, u_1]$ denote the desired fairness level; i.e., we wish to select $\lambda$ such that the solution of (2), $\mathbf{x}^*(\lambda)$, is the densest subgraph that satisfies $r_1(\mathbf{x}^*(\lambda)) = \alpha$. To this end, we will exploit the fact that the function $r_1(\mathbf{x}^*(\lambda))$ is *non-decreasing in* $\lambda$. Consequently, if we define the function

$$\psi_1(\lambda) := r_1(\mathbf{x}^*(\lambda)) - \alpha, \tag{5}$$

we can employ bisection on $\lambda$ to solve the equation $\psi_1(\lambda) = 0$, in order to meet the target fairness level $\alpha \in [l_1, u_1]$. The lower and upper limits of the bisection interval correspond to 0 and $\lambda_{\max}$ respectively, for which $\psi_1(0) = l_1 - \alpha \leq 0$ and $\psi_1(\lambda_{\max}) = 1 - \alpha \geq 0$. Moreover, note that each choice of $\lambda \in [0, \lambda_{\max}]$ corresponds to solving an instance of (2), which, as pointed out before, is tractable.

**FADSG-II:** A similar technique can be applied to select $\lambda$ in problem (4) as well. Let $\mathbf{x}^*(0)$ denote the solution of problem (4), which again corresponds to DSG, and define $u_2 := r_2(\mathbf{x}^*(0)) \geq 0$. In this case, $\mathbf{x}^*(\lambda_{\max})$ corresponds to the entire protected subset, for which $l_2 := r_2(\mathbf{x}^*(\lambda_{\max})) = 0$. Since $r_2(\mathbf{x}^*(\lambda))$ is *non-increasing* in $\lambda$, given a desired value of $r_2(\cdot)$ defined as $\delta \in [l_2, u_2]$, we can solve the equation

$$\psi_2(\lambda) := r_2(\mathbf{x}^*(\lambda)) - \delta = 0 \tag{6}$$

via bisection on $\lambda$. The initial bisection interval is $[0, \lambda_{\max}]$, with $\psi_2(0) = u_2 - \delta \geq 0$ and $\psi_2(\lambda_{\max}) = -\delta \leq 0$. For each $\lambda \in [0, \lambda_{\max}]$, an instance of problem (4) has to be solved, which is also tractable.

Unlike FADSG-I, in this case it is more difficult to set $\delta$ to a pre-determined value as it does not admit a straight-forward interpretation of corresponding to a particular target fairness level. An important exception is the case where $\delta = 1$, which is equivalent to the requirement that 50% of the protected subset is represented in the solution (see Lemma 4). Thus, the bisection-search strategy can be applied to determine the value of $\lambda$ that will extract the densest subgraph that contains 50% of the protected vertices.

A subtle point is that for the bisection framework to provably terminate, the functions $\psi_1(\lambda)$ and $\psi_2(\lambda)$ should be continuous for every target fairness level, and for every graph. Although we cannot show this at present, we observed in our experiments that depending on the nature of the underlying Pareto-optimal frontier, some target fairness levels may be more difficult to attain than others. However, in general, the overall procedure works very well on real datasets.

## 5 Quantifying the Price of Fairness

FADSG-I and II trade off subgraph density in order to promote the inclusion of protected vertices. Consequently, the density of the fair subgraphs obtained via (2) and (4) is no larger than the density of their non-fair counterpart DSG; i.e., a loss in subgraph density is the price paid for ensuring fairness. The loss incurred in attaining a target fairness level $\phi$, either $\alpha$ for FADSG-I or $\delta$ for FADSG-II, can be quantified using the *price of fairness*.

For a given graph $\mathcal{G}$, the *price of fairness* (PoF) for a fixed value of $\phi$ is

$$\mathrm{PoF}(\mathcal{G},\phi) := \frac{\rho_\mathcal{G}^* - \rho_\mathcal{G}^*(\phi)}{\rho_\mathcal{G}^*} := 1 - \frac{\rho_\mathcal{G}^*(\phi)}{\rho_\mathcal{G}^*}, \tag{7}$$

where $\rho_\mathcal{G}^*(\phi)$ is the density of the densest subgraph in $\mathcal{G}$ attaining a target fairness level $\phi$, and $\rho_\mathcal{G}^*$ is the density of the solution of DSG. Simply stated, $\mathrm{PoF}(\mathcal{G},\phi)$ is the ratio of the loss in subgraph density suffered in meeting the fairness requirement $\phi$ and the maximum density $\rho_\mathcal{G}^*$ in the absence of fairness considerations. Let $\phi_0$ denote the fairness level of the solution of DSG (obtained with explicitly imposing fairness), then $\rho_\mathcal{G}^*(\phi_0) = \rho_\mathcal{G}^*$, and thus $\mathrm{PoF}(\mathcal{G},\phi_0) = 0$. As $\phi$ diverges from $\phi_0$, the resulting density $\rho_\mathcal{G}^*(\phi)$ decreases, and consequently $\mathrm{PoF}(\mathcal{G},\phi)$ approaches 1. For a desired value of $\phi$, the closer $\mathrm{PoF}(\mathcal{G},\phi)$ is to 1, the greater is the price paid (in terms of density) for extracting the desired fair dense subgraph.

Similar definitions have been considered for analyzing fairness in influence maximization (Tsang et al., 2019) and graph covering (Rahmattalabi et al., 2019). To the best of our knowledge, however, we are the first to study such a notion in the context of detecting fair dense subgraphs. Analyzing the price of fairness is inherently complex due to multiple factors like the type of graph considered, the protected subgraph, and the desired level of fairness.

**The "Lollipop" Graph:** We base our study on a difficult example which exemplifies the underlying tension between density and fairness. Consider the "lollipop" graph (denoted as $\mathcal{L}_n$). In this construction, the unprotected vertices constitute a clique of size $\sqrt{n}$ ($\mathcal{C}_{\sqrt{n}}$), while the protected vertices form a path graph on $n - \sqrt{n}$ vertices ($\mathcal{P}_{n-\sqrt{n}}$), with one end connected to $\mathcal{C}_{\sqrt{n}}$ (so that the overall graph is connected). We point out that the "lollipop" graph mimics the properties of real-world graphs in the following manner: (i) the graph $\mathcal{L}_n$ has $O(n)$ edges, and is hence sparse, (ii) the majority of vertices ($n - \sqrt{n}$ to be precise) have low degree ($\leq 2$), while the remaining vertices have high degree (at least $\sqrt{n} - 1$). Intuitively, we expect the solution of DSG for the "lollipop" graph to be the clique $\mathcal{C}_{\sqrt{n}}$, as it corresponds to a region of high density. Since this solution does not contain any protected vertices, from a fairness perspective, it is the worst possible solution. In addition, as the protected subset $\mathcal{P}_{n-\sqrt{n}}$ is minimally connected, and thus has low density, we expect that improving the

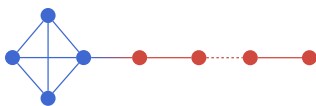

Figure 3: The "lollipop" graph for $n = 16$. The unprotected set (blue) forms a clique on 4 vertices whereas the protected set (red) forms a path graph on 12 vertices.

representation of protected vertices will cost a large decrease in density, reflected by a large price of fairness.

**FADSG-I:** Let us analyse how the PoF is affected by altering the balance between the majority and minority vertices in the desired solution. Consider the case where the desired proportion of unprotected and protected vertices in the extracted subgraph is $1 : \gamma$, with $\gamma \in \{1, 2, \cdots, n - \sqrt{n}\}$, and the desired balance is $\alpha_\gamma := \gamma/(1 + \gamma)$. Note that the case of $\gamma = 1$ corresponds to the perfect balance scenario considered in Anagnostopoulos et al. (2020; 2024). Then, the PoF can be expressed as follows (see Appendix D.1 for proof).

**Proposition 7.**

$$PoF(\mathcal{L}_n, \alpha_\gamma) \begin{cases} = \alpha_\gamma \left[ 1 - \frac{2}{\sqrt{n}-1} \right], & \alpha_\gamma \in \left\{ \frac{1}{2}, \cdots, 1 - \frac{1}{\sqrt{n}} \right\} \\ > 1 - \left[ \frac{1}{(\sqrt{n})} + \frac{2}{\sqrt{n}-1} \right], & \alpha_\gamma \in \left\{ 1 - \frac{1}{\sqrt{n}+1}, \cdots, 1 - \frac{1}{n-\sqrt{n}+1} \right\} \end{cases} \tag{8}$$

Since the target fairness level $\alpha_\gamma$ increases with $\gamma$, the above result reveals that the PoF increases with $\gamma$, as expected. Perhaps surprisingly, even when $\alpha_\gamma$ is large but (strictly) lesser than 1, it is still possible that the PoF is at least $1 - O(1/\sqrt{n})$, which can again be made arbitrarily close to 1. For smaller values of $\alpha_\gamma$, we have that $\lim_{n\to\infty} \mathrm{PoF}(\mathcal{L}_n, \alpha_\gamma) = \alpha_\gamma$. In this case, the PoF is at most the desired fairness level $\alpha_\gamma$.

**FADSG-II:** Regarding FADSG-II, we can show that for the lollipop graph, there exists a one-to-one correspondence between the target fairness parameters $\delta$ of FADSG-II and $\alpha$ of FADSG-I such that solutions of the two problems coincide (refer to Appendix D.2 for proof). Hence the analysis remains the same as that of FADSG-I.

# 6 Algorithmic Approach

Although FADSG-I and II can be optimally solved using maximum flows, we refrain from using such an approach. The main reason is that for a target fairness level, we have to solve multiple instances of each problem within a binary search framework to determine the appropriate regularization parameter $\lambda$. Since solving each maximum flow problem incurs $\Omega(n^2)$ complexity (even for sparse graphs), the overall procedure for computing an exact solution can be computationally prohibitive. Thus, we adopt an approximation approach that seeks to quickly compute high quality sub-optimal solutions for a given instance of FADSG-I and II (i.e., for a fixed $\lambda$). The key observation is that both of these problems correspond to special cases of a general problem, the DENSEST SUPERMODULAR SUBSET (DSS) problem (Chekuri et al., 2022). Given a supermodular function $F : 2^{\mathcal{V}} \to \mathbb{R}$, the problem seeks a subset that maximizes the ratio $F(\mathcal{S})/|\mathcal{S}|$. For this problem class, several algorithmic strategies are available. One option is to employ SUPER-GREEDY++ (Chekuri et al., 2022), a multi-stage greedy algorithm which generalizes the prior-art of Asahiro et al. (2000); Boob et al. (2020); Veldt et al. (2021). If $F(\cdot)$ is normalized, non-negative, and monotone, for a given $\epsilon \in (0, 1)$, the algorithm outputs an $(1 - \epsilon)$-approximation of the optimal objective value of DSS in $O\left(1/\epsilon^2\right)$ iterations. The peeling process dominates the computational complexity of each iteration and it can be implemented to run in $O(m + n \log n)$, by handling the vertices of $\mathcal{S}$ using a Fibonacci heap (Fredman & Tarjan, 1987) with key values being $\ell_v + F(v|\mathcal{S} - v)$, similar to Miyauchi et al. (2023). Hence, SUPER-GREEDY++ can run in $O((m + n \log n)/\epsilon^2)$. Its main appeal is that it can be efficiently implemented, and exhibits strong empirical performance (Boob et al., 2020; Nguyen & Ene, 2024). Additionally, it is straightforward to implement it using the objective functions of FADSG-I and II (for more details, see [Algorithm 1, Appendix E]).

An alternative is to consider the iterative convex-optimization algorithms developed in Danisch et al. (2017); Harb et al. (2022); Nguyen & Ene (2024). These methods are centered around solving the dual LP relaxation of DSG (Charikar, 2000). However, modifying and implementing these algorithms (which are tailored for DSG) for our problems poses a greater challenge relative to SUPER-GREEDY++. FADSG-I and II have to be reformulated as a constrained convex optimization problem via an extension of the dual linear programming (LP) relaxation of DSG proposed in Charikar (2000). Moreover, since the algorithms of Danisch et al. (2017); Harb et al. (2022); Nguyen & Ene (2024) are tailored to exploit the structure of the dual LP relaxation based on DSG, it is not clear to what extent they would need to be modified in order to be applicable for our problems, and whether the requisite changes would still result in low-complexity updates. Another practical drawback of these continuous methods (even when applied to plain DSG) is that a subroutine known as fractional peeling (Harb et al., 2022) has to be applied on each of the generated iterates if one desires to extract an estimate of the densest supermodular subset (which corresponds to the densest subgraph for DSG) at each iteration. However, performing a single round of fractional peeling incurs a complexity of $O(m + n \log n)$, which has the same complexity order as a single iteration of SUPER-GREEDY++. Consequently, determining the point at which the estimate of the DSS has converged from the continuous iterates can prove to be computationally expensive, unless the fractional peeling subroutine is sparingly used. Even then, this can come at the cost of executing the convex optimization algorithms for a larger number of iterations than required for the sequence of DSS estimates to converge. We point out that this is a non-issue in SUPER-GREEDY++, as its combinatorial nature makes it easy to keep track of the best estimate of the DSS at each iteration. Owing to these reasons, we elect to choose SUPER-GREEDY++ as the algorithmic primitive for FADSG-I and II.

Note that the objective function of FADSG-I can be expressed in the form of the DSS problem, with a non-negative, monotone supermodular numerator. Hence, the SUPER-GREEDY++ algorithm can be applied directly to problem (2) to obtain a high-quality approximate solution. For FADSG-II, its objective function can be expressed as well as the ratio $F(\mathcal{S})/|\mathcal{S}|$, where the numerator is supermodular, making problem (4) a special case of DSS. However, since the numerator may not always be non-negative for every $\lambda > 0$, SUPER-GREEDY++ does not guarantee an $(1 - \epsilon)$-approximation in all cases (see Huang et al. (2024) for a counter-example). Nonetheless, SUPER-GREEDY++ can still be applied on problem (4) and our experiments indicate that it can serve as an effective heuristic. For details regarding implementation, refer to Appendix E.

It is worth pointing out that although we employ approximation algorithms for our problem formulations, there is a difference compared to that of the prior art Anagnostopoulos et al. (2024; 2020); Miyauchi et al. (2023). More specifically, we employ approximation as a means of improving scalability, as opposed to dealing with intractability.

Table 1: Summary of dataset statistics: number of vertices ($n$), number of edges ($m$), and size of protected subset ($n_p$). The datasets contain multiple graphs and the statistics represent mean values with standard deviations. The full list of datasets can be found in Appendix F.

| DATASET | $n$ | $m$ | $n_p$ |
|---|---|---|---|
| AMAZON (10) | $3699 \pm 2764$ | $22859 \pm 18052$ | $1927 \pm 2537$ |
| LASTFM (19) | 7624 | 27806 | $424 \pm 454$ |
| TWITCH (6) | $5686 \pm 2393$ | $71519 \pm 45827$ | $2523 \pm 1787$ |

## 7 Experimental Evaluation

**Experimental Setup:** We consider various real-world datasets, with Table 1 summarizing their statistics (for more information, see Appendix F). The implementation of our approach is publicly available on Github[1].

The following baselines were selected as performance benchmarks:

- *2-DFSG* (Anagnostopoulos et al., 2020; 2024). The algorithm starts from the solution of DSG and subsequently introduces vertices from the underrepresented group in an *arbitrary* fashion, until perfect balance is reached.

- *PS, FPS* (Anagnostopoulos et al., 2020; 2024). Paired Sweep (PS) and Fair Paired Sweep (FPS) are spectral algorithms. They categorize the elements based on their group and sort them in a manner reminiscent of spectral clustering.

- *DDSP* (Miyauchi et al., 2023). The algorithm initially computes a constant-factor approximate solution to Dal$k$S with $k = \lceil 1/\alpha \rceil$, with $\alpha$ being the maximum desired fraction of monochromatic vertices. Then, the algorithm makes the solution feasible by adding an *arbitrary* vertex from the least represented group, similar to 2-DFSG.

### 7.1 Evaluation of FADSG-I

**Comparison with prior art:** In order to compare FADSG-I with the baselines, we set the target fairness level $\alpha = 0.5$. Therefore, our formulation aims to extract a dense subgraph with perfect balance. We point out that we exclude datasets for which the protected vertices in the DSG solution are more than the unprotected, as they already meet and exceed the target $\alpha = 0.5$. Figures 2 and 4 illustrate a comparative analysis with prior art. The top sections showcase the price of fairness (PoF) for each algorithm across various datasets, while the bottom sections depict the fraction of protected vertices in the solutions. As the results indicate, our proposed formulation consistently yields the lowest PoF values across a broad spectrum of scenarios.[2]

It can be observed from Figures 2 and 4 that for certain datasets, FADSG-I does not find a perfectly balanced subgraph (see Figure 4, AMAZON HPC & OP). To obtain a better understanding, we plot the regularization path of $r_1(\mathcal{S})$ for these datasets in Figure 5. A noteworthy observation is the emergence of "fairness plateaus" where the fairness level remains constant across intervals of $\lambda$. Since FADSG-I is tractable, these figures correspond to the *actual* Pareto frontier which characterizes the trade-off between fairness and density. This phenomenon suggests that the graph's inherent structure leads to certain fairness levels that arise naturally. For instances where the data's underlying structure does not inherently support a perfect division (i.e., AMAZON HPC & OP), FADSG-I does not enforce it. Figure 6 depicts the outcomes in datasets where there exists an extreme imbalance in the composition of the protected and unprotected subgroups, i.e., we have $n_p \ll n$ (see Table 4, Appendix F). In such cases, DDSP consistently fails to produce balanced/fair solutions. In fact, we observe an extreme gap to the desired fairness level of 0.5, as well as a PoF that is always worse than that of FADSG-I. The other prior methods always achieve a perfect balance, but with a significantly higher PoF compared to our algorithm. This is an important feature of FADSG-I, as it can provide a near optimal solution, when the nature of the graph allows it, while almost always paying the least in terms of PoF.

---

[1]https://github.com/ekariotakis/fadsg_tmlr2025
[2]Note that, in all experiments, in order to compute $\rho_{\mathcal{G}}^*$, we employ the exact max flow algorithm (Goldberg, 1984).

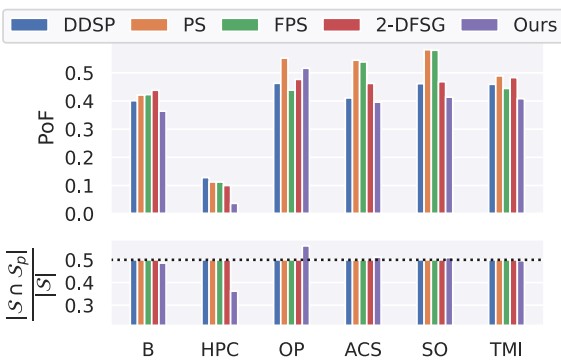

Figure 4: Comparing induced perfectly balanced fair subsets of prior art and FADSG-I (Ours), on different AMAZON datasets. Top: PoF. Bottom: Fraction of protected vertices in induced subsets, $r_1(\mathcal{S})$.

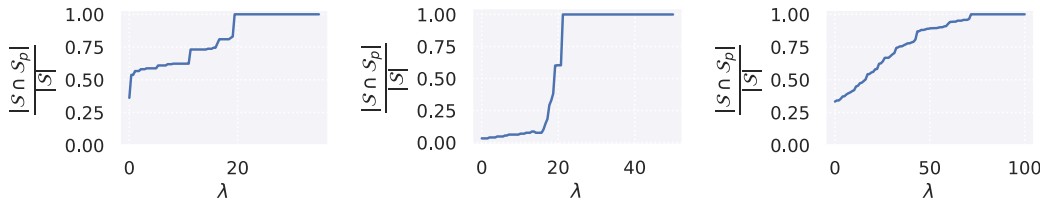

Figure 5: The fraction of protected vertices in the induced subset, $r_1(\mathcal{S})$, as function of $\lambda$, for FADSG-I. Left: AMAZON HPC - Middle: AMAZON OP - Right: TWITCH PTBR.

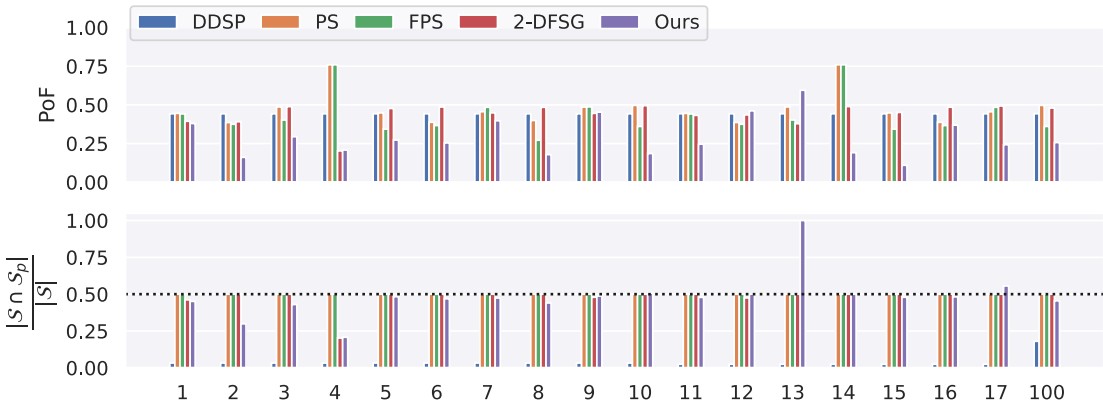

Figure 6: Comparing induced perfectly balanced fair subsets of prior art and FADSG-I (Ours), on different LASTFM datasets. Top: PoF. Bottom: Fraction of protected vertices in induced subsets, $r_1(\mathcal{S})$.

**Various target fairness levels:** Next, we investigate the ability of FADSG-I to extract subgraphs corresponding to various fairness levels, contrary to PS, FPS, or 2-DFSG. Although DDSP allows adjusting group representation levels, it can't guarantee *a priori* which group (i.e., protected or unprotected) will meet them. Returning to Figure 5, it can be observed that the fairness levels are non-decreasing with $\lambda$. Notably, this value can only surpass or equal that of the classic DSG solution ($\lambda = 0$), aligning with our objective of enhancing fairness. The achievable fairness levels are contingent upon the underlying nature of the dataset,

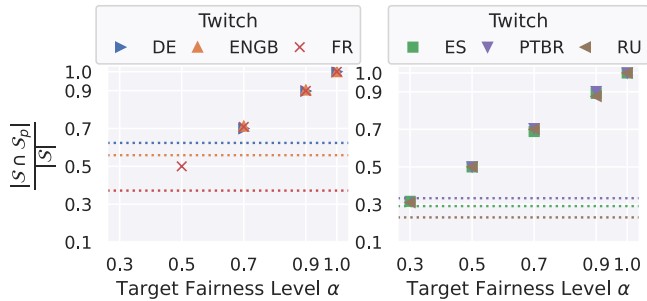

Figure 7: Fraction of protected vertices in each induced subgraph, $r_1(\mathcal{S})$, of FADSG-I, for various values of target fairness level $\alpha$, for the TWITCH datasets. (Dotted lines: fairness value in the solution of classic DSG that we want to enhance.)

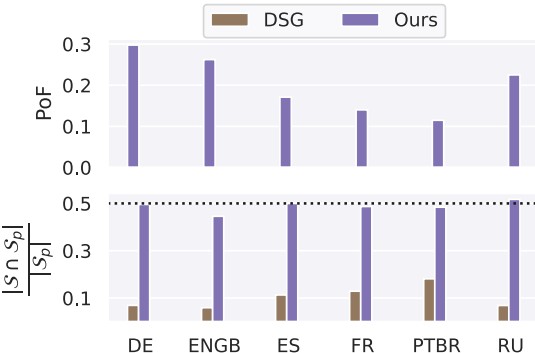

Figure 8: TWITCH datasets. Top: PoF of FADSG-II (Ours) for $\delta = 1$. Bottom: Comparing Ours ($\delta = 1$) with DSG. Fraction of the protected subset that belongs in the solutions.

with datasets like TWITCH PTBR exhibiting smoother curves (Figure 5 - Right), resulting in naturally varying degrees of fairness. In contrast, certain datasets, such as AMAZON HPC & OP (Figure 5), display plateaus that correspond to specific fairness levels induced in the final solution. Figure 7 demonstrates the flexibility of FADSG-I in extracting fair dense subgraphs that attain varying target fairness levels $\alpha$, which correspond to the fraction of protected vertices in the induced subgraph. Starting from the fairness level attained by the DSG solution, we can introduce more fairness to our solution by increasing the desired value of $\alpha$.

## 7.2 Evaluation of FADSG-II

Direct comparison with prior art is not feasible here, since FADSG-II utilizes a different fairness criterion, as detailed in Section 3. In Figure 8, we compare this formulation with DSG. The top section depicts the PoF of the solution obtained from FADSG-II, and the bottom one illustrates the fraction of all of the protected vertices in the solutions of DSG and FADSG-II. For the TWITCH datasets, DSG's solution includes only about 1/5 of the protected vertices. This is clearly due to the fact that DSG does not take into account the protected attributes of the vertices. On the other hand, FADSG-II for $\delta = 1$, yields solutions with nearly half of the protected vertices, while achieving a relatively low PoF. Similar to FADSG-I, there are instances where the solution does not precisely align with the desired fairness level. This can be attributed to the structure of each graph, which may not inherently exhibit the specific fraction we are aiming for (see Appendix H for more details).

## 8 Conclusions

We addressed the fairness-aware densest subgraph problem by introducing two tractable formulations, each incorporating a different regularizer in the objective function, to provide two different notions of fairness. The inclusion of fairness introduces the inherent fairness-density trade-off. In order to quantify this trade-off we proposed a measure of the price of fairness, and we analyzed it based on an illuminating example – which revealed the underlying tension between density and fairness. To assess the effectiveness of the proposed solutions, we employed a peeling-based approximation algorithm, and compared their performance against the prior art using real-world datasets. Our findings revealed that FADSG-I generally incurs a lower price of fairness (PoF) in achieving perfect balanced subgraphs, sometimes even less than half of that of prior art. FADSG-I can also induce a perfectly balanced subgraph in some extreme scenarios where existing approaches fall short, and it is capable of providing various fairness levels, other than perfect balance. Furthermore, FADSG-II promotes an alternative notion of inclusion of protected vertices, often achieving this with a relatively low PoF. Together, these formulations offer flexible solutions for balancing fairness and density in subgraph discovery. Extending our approach to handle multiple vertex groups using regularizers which are designed to ensure that no protected group is inadequately represented constitutes a direction of future work.

## Broader Impact

The authors posit this work on the assumption that practitioners will appropriately select the protected subset and publicly disclose it along with the results of their analysis.

## Acknowledgments

Supported in part by the National Science Foundation, USA under grant IIS-1908070 and the KU Leuven Special Research Fund BOF/STG-22-040.

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

## A   Lemmas

### A.1   Proof of Lemma 4

*Proof.* We express $r_2(.)$ in set notation as

$$r_2(\mathcal{S}) = \frac{|\mathcal{S}| + |\mathcal{S}_p| - 2|\mathcal{S} \cap \mathcal{S}_p|}{|\mathcal{S}|}. \tag{9}$$

The claim then follows by noting that $r_2(\mathcal{S}) = 1 \Leftrightarrow |\mathcal{S}_p| - 2|\mathcal{S} \cap \mathcal{S}_p| = 0$. $\quad\square$

### A.2   Proof and Intuition of Lemma 5

*Proof.* Note that $r_2(\cdot)$ can be expressed in terms of $r_1(\cdot)$ as follows

$$r_2(\mathcal{S}) = 1 + \frac{n_p}{|\mathcal{S}|} - 2r_1(\mathcal{S}). \tag{10}$$

• For the first inequality: Since $|\mathcal{S}_p| \geq |\mathcal{S}_p \cap \mathcal{S}| = r_1(\mathcal{S})|\mathcal{S}|$, it implies that $\frac{|\mathcal{S}_p|}{|\mathcal{S}|} \geq r_1(\mathcal{S})$. Hence, we obtain

$$r_2(\mathcal{S}) \geq 1 - r_1(\mathcal{S}). \tag{11}$$

• For the second inequality: We have that $\frac{n_p}{|\mathcal{S}|} \leq \frac{n_p}{2}$, since each solution has at least one edge. Hence, we obtain

$$2r_1(\mathcal{S}) + r_2(\mathcal{S}) \leq 1 + \frac{n_p}{2}. \tag{12}$$

$\square$

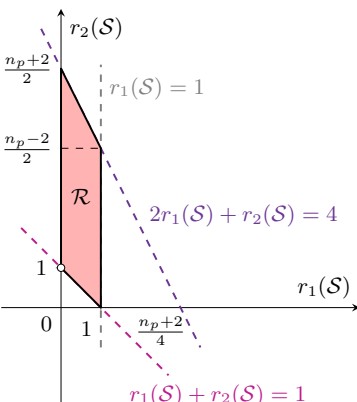

Figure 9: The joint fairness region of the regularizers $r_1(\mathcal{S})$ and $r_2(\mathcal{S})$, for a graph with $n_p$ protected vertices.

Furthermore, we define the joint fairness region characterized by the above inequalities as the following polytope:

$$\mathcal{R} := \left\{ (r_1(\mathcal{S}), r_2(\mathcal{S})) : 0 \leq r_1(\mathcal{S}) \leq 1, 0 \leq r_2(\mathcal{S}) \leq 1 + \frac{n_p}{2}, r_1(\mathcal{S}) + r_2(\mathcal{S}) \geq 1, 2r_1(\mathcal{S}) + r_2(\mathcal{S}) \leq 1 + \frac{n_p}{2} \right\} \tag{13}$$
$$\setminus \{ r_1(\mathcal{S}) = 0 \wedge r_2(\mathcal{S}) = 1 \}$$

Note that the point $(0, 1)$ is excluded from $\mathcal{R}$, since if $r_1(\mathcal{S}) = 0 \Leftrightarrow |\mathcal{S} \cap \mathcal{S}_p| = 0$ and $r_2(\mathcal{S}) = 1 \Leftrightarrow |\mathcal{S} \cap \mathcal{S}_p| = \frac{1}{2}|\mathcal{S}_p| \Rightarrow |\mathcal{S}_p| = 0$, which is a contradiction since we assume that there always exists at least one protected vertex in the graph. We can approximate this point when $n_p \ll n$. In this case, when $r_1(\mathcal{S}) \to 0$, it can still hold that half of the protected vertices belong in the solution, i.e., $r_2(\mathcal{S}) = 1$.

Let us consider the three remaining corner-points of $\mathcal{R}$.

- (Bottom right:) We can attain $(1, 0)$ when $r_2(\mathcal{S}) = 0$, meaning that the solution is the protected set $\mathcal{S}_p$, hence $r_1(\mathcal{S}) = \frac{|\mathcal{S} \cap \mathcal{S}_p|}{|\mathcal{S}|} = 1$.

- (Top left:) We can attain $(0, \frac{n_p+2}{2})$ only when the solution contains just one edge that connects two *unprotected* vertices. This holds because when $r_1(\mathcal{S}) = 0 \Leftrightarrow |\mathcal{S} \cap \mathcal{S}_p| = 0 \Rightarrow r_2(\mathcal{S}) = \frac{n_p}{2} + 1$, which is equal to the desired value only for $|\mathcal{S}| = 2$.

- (Top right:) We can attain $(1, \frac{n_p-2}{2})$ only when the solution contains just one edge that connects two *protected* vertices. This holds because when $r_1(\mathcal{S}) = 1 \Leftrightarrow |\mathcal{S} \cap \mathcal{S}_p| = |\mathcal{S}| \Rightarrow r_2(\mathcal{S}) = \frac{n_p}{2} - 1$, which is equal to the desired value only for $|\mathcal{S}| = 2 \Leftrightarrow |\mathcal{S} \cap \mathcal{S}_p| = 2$.

# B Tractability details

First, consider problem (2). Define the functions $p_1(\mathbf{x}, \lambda) := \mathbf{x}^\top \mathbf{A}\mathbf{x} + \lambda \cdot \mathbf{e}_p^\top \mathbf{x}$, and $q(\mathbf{x}) := \mathbf{e}^\top \mathbf{x}$. Thus, the objective function can be expressed as $g(\mathbf{x}, \lambda) = p_1(\mathbf{x}, \lambda)/q(\mathbf{x})$. Since $p_1(\cdot, \lambda)$ is the sum of a non-negative, monotone supermodular function and a non-negative modular function, the result is also non-negative, monotone supermodular (for every $\lambda \geq 0$). This implies that FADSG-I corresponds to maximizing the ratio of a monotone, non-negative supermodular function and a modular function. The solution to such a problem can be determined in polynomial-time by solving a finite number of supermodular maximization (SFM) sub-problems within a binary-search framework (Fujishige, 2005; Bai et al., 2016). Since SFM incurs polynomial-time complexity (Grötschel et al., 2012), the claim follows. Furthermore, the quadratic form of $p_1(\cdot, \lambda)$ can be exploited to show that each sub-problem is equivalent to solving a minimum-cut problem (Goldberg, 1984; Kolmogorov & Zabih, 2002), which can be accomplished via maximum-flow, as we show in Appendix B.1.1. Each maximum-flow problem incurs complexity $O(nm)$ (Orlin, 2013) and an upper-bound on the maximum number of binary-search steps required is $O(\log(n + \lambda) + \log(n))$ (see Appendix B.2).

Next, consider problem (4), and define the function $p_2(\mathbf{x}, \lambda) := \mathbf{x}^\top \mathbf{A}\mathbf{x} - \lambda \cdot (\mathbf{e} - 2\mathbf{e}_p)^\top \mathbf{x} - \lambda \cdot n_p$. Then, the objective function of FADSG-II can be expressed as the ratio $h(\mathbf{x}, \lambda) = p_2(\mathbf{x}, \lambda)/q(\mathbf{x})$. While $p_2(\cdot, \lambda)$ is a supermodular function (being the sum of a supermodular and a modular function), it is not guaranteed to be non-negative for every choice of $\lambda > 0$. Additionally, it is not normalized either, since $p_2(\mathbf{0}, \lambda) = -\lambda \cdot n_p$. Nevertheless, we can still show that (4) can be solved via a sequence of maximum-flow problems using a small modification of the binary-search strategy, as we explain in Appendix B.1.2. Thus, FADSG-II can also be solved exactly in polynomial-time.

## B.1 Exact solutions via max-flow

### B.1.1 Applying max-flow on FADSG-I

Based on the interpretation of Lanciano et al. (2024, Algorithm 1) for Goldberg's max-flow based algorithm (Goldberg, 1984), we only need to make a small modification in order to apply it to FADSG-I.

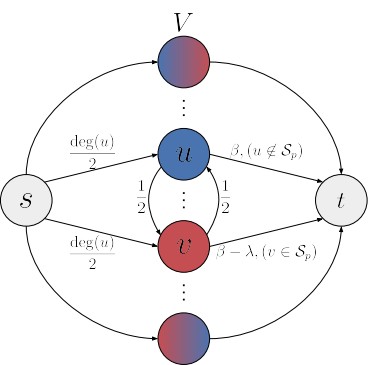

The algorithm maintains an estimate of the unknown optimal value by tracking upper and lower bounds, which it progressively tightens through binary search. To update them, the algorithm employs a max-flow (equivalently min-cut) computation. Let $\beta \geq 0$ be the midpoint of the bounds kept in the current iteration of the max-flow based algorithm. For $\mathcal{G} = (\mathcal{V}, \mathcal{E})$, $\mathcal{S}_p$ the protected set and $\lambda$ the regularization parameter of FADSG-I (constant), the algorithm constructs the following edge-weighted directed graph $(\mathcal{U}, \mathcal{A}, w_\beta^I)$ (Figure 10), with $\mathcal{U} = \mathcal{V} \cup \{s, t\}$, $\mathcal{A} = \mathcal{A}_s \cup \mathcal{A}_\mathcal{E} \cup \mathcal{A}_t^p \cup \mathcal{A}_t^{\bar{p}}$, where $\mathcal{A}_s = \{(s, v) | v \in \mathcal{V}\}$, $\mathcal{A}_\mathcal{E} = \{(u, v), (v, u) | \{u, v\} \in \mathcal{E}\}$, $\mathcal{A}_t^p = \{(u, t) | v \in \mathcal{S}_p\}$ and $\mathcal{A}_t^{\bar{p}} = \{(u, t) | v \in \bar{\mathcal{S}}_p\}$ and $w_\beta^I : \mathcal{A} \to \mathbb{R}_+$ such that

$$w_\beta^I(e) = \begin{cases} \deg(v) & e = (s, v) \in \mathcal{A}_s \\ 1 & e \in \mathcal{A}_\mathcal{E} \\ \beta - \lambda & e \in \mathcal{A}_t^p \\ \beta & e \in \mathcal{A}_t^{\bar{p}} \end{cases}. \quad (14)$$

Figure 10: An edge-weighted directed graph $(\mathcal{U}, \mathcal{A}, w_\beta)$ constructed from $\mathcal{G}$, $\beta$ and $\lambda$.

An $s - t$ cut of $(\mathcal{U}, \mathcal{A}, w_\beta)$ is a partition of $(\mathcal{X}, \mathcal{Y})$ of $\mathcal{U}$, such that $s \in \mathcal{X}$ and $t \in \mathcal{Y}$. Then define the cost of an $s - t$ cut $(\mathcal{X}, \mathcal{Y})$, as $\text{cost}(\mathcal{X}, \mathcal{Y}) := \sum_{(u,v) \in \mathcal{A}: u \in \mathcal{X}, v \in \mathcal{Y}} w_\beta(u, v)$.

**Lemma 8.** *Let $(\mathcal{X}, \mathcal{Y})$ be an $s - t$ cut of the edge-weighted directed graph $(\mathcal{U}, \mathcal{A}, w_\beta^I)$, and $\mathcal{S} = \mathcal{X} \setminus \{s\}$. Then it holds that $\text{cost}(\mathcal{X}, \mathcal{Y}) = 2m + \beta|\mathcal{S}| - p_1(\mathcal{S}, \lambda)$, where $p_1(\mathcal{S}, \lambda) = 2e(\mathcal{S}) + \lambda|\mathcal{S} \cap \bar{\mathcal{S}}_p|$.*

*Proof.* Any edge from $\mathcal{X}$ to $\mathcal{Y}$ is contained in exactly one of the following sets: $\{(s,v) \in \mathcal{A}_s | v \in \mathcal{V} \setminus \mathcal{S}\}$, $\{(u,v) \in \mathcal{A}_\mathcal{E} | u \in \mathcal{S}, v \in \mathcal{V} \setminus \mathcal{S}\}$, $\{(v,t) \in \mathcal{A}_t^p | v \in \mathcal{S}\}$, $\{(v,t) \in \mathcal{A}_t^{\bar{p}} | v \in \mathcal{S}\}$. Therefore the cost of $(\mathcal{X}, \mathcal{Y})$ is

$$
\begin{aligned}
\text{cost}(\mathcal{X}, \mathcal{Y}) &= \sum_{(s,v) \in \mathcal{A}_s : v \in \mathcal{V} \setminus \mathcal{S}} w_\beta^I(s,v) + \sum_{(u,v) \in \mathcal{A}_\mathcal{E} : u \in \mathcal{S}, v \in \mathcal{V} \setminus \mathcal{S}} w_\beta^I(u,v) + \sum_{(u,t) \in \mathcal{A}_t^p : v \in \mathcal{S}} w_\beta^I(v,t) + \sum_{(u,t) \in \mathcal{A}_t^{\bar{p}} : v \in \mathcal{S}} w_\beta^I(v,t) \\
&= \sum_{v \in \mathcal{V} \setminus \mathcal{S}} \deg(v) + |\{(u,v) \in \mathcal{E} | u \in \mathcal{S}, v \in \mathcal{V} \setminus \mathcal{S}\}| + (\beta - \lambda)|\mathcal{S} \cap \mathcal{S}_p| + \beta(|\mathcal{S}| - |\mathcal{S} \cap \mathcal{S}_p|) \\
&= 2m - 2e(\mathcal{S}) + \beta|\mathcal{S}| - \lambda|\mathcal{S} \cap \mathcal{S}_p| = 2m + \beta|\mathcal{S}| - 2e(\mathcal{S}) - \lambda|\mathcal{S} \cap \mathcal{S}_p| \\
&= 2m + \beta|\mathcal{S}| - p_1(\mathcal{S}, \lambda).
\end{aligned}
\tag{15}
$$

$\square$

Let $(\mathcal{X}, \mathcal{Y})$ be the minimum $s - t$ cut (or max-flow) of $(\mathcal{U}, \mathcal{A}, w_\beta^I)$. If $\mathcal{X} \neq \{s\}$, then $\mathcal{S} = \mathcal{X} \setminus \{s\}$ is a vertex subset that satisfies $\beta|\mathcal{S}| - p_1(\mathcal{S}, \lambda) \leq 0 \Leftrightarrow \frac{p_1(\mathcal{S}, \lambda)}{|\mathcal{S}|} \geq \beta$, and we can update the lower bound by $\beta$. If $\mathcal{X} = \{s\}$, then there is no $\mathcal{S} \subset \mathcal{V}$ that satisfies $\frac{p_1(\mathcal{S}, \lambda)}{|\mathcal{S}|} > \beta$, and the upper bound can be replaced by $\beta$. Therefore, we can apply the same procedure as that of Lanciano et al. (2024, Algorithm 1), with a small change at the initial upper bound $\beta_{\text{ub}}^{(0)} \leftarrow m$.

### B.1.2 Applying max-flow on FADSG-II

We follow the same procedure as we did in Appendix B.1.1, but with a slightly different edge-weighted graph. For $\mathcal{G} = (\mathcal{V}, \mathcal{E})$, $\mathcal{S}_p$ the protected set and $\lambda$ the regularization parameter of FADSG-II (constant), the algorithm constructs the following edge-weighted directed graph $(\mathcal{U}, \mathcal{A}, w_\beta^{II})$, with $\mathcal{U} = \mathcal{V} \cup \{s,t\}$, $\mathcal{A} = \mathcal{A}_s \cup \mathcal{A}_\mathcal{E} \cup \mathcal{A}_t^p \cup \mathcal{A}_t^{\bar{p}}$, where $\mathcal{A}_s = \{(s,v) | v \in \mathcal{V}\}$, $\mathcal{A}_\mathcal{E} = \{(u,v), (v,u) | \{u,v\} \in \mathcal{E}\}$, $\mathcal{A}_t^p = \{(u,t) | v \in \mathcal{S}_p\}$ and $\mathcal{A}_t^{\bar{p}} = \{(u,t) | v \in \bar{\mathcal{S}}_p\}$ and $w_\beta^{II} : \mathcal{A} \to \mathbb{R}_+$ such that

$$
w_\beta^{II}(e) = \begin{cases}
\deg(v) & e = (s,v) \in \mathcal{A}_s \\
1 & e \in \mathcal{A}_\mathcal{E} \\
\beta - \lambda & e \in \mathcal{A}_t^p \\
\beta + \lambda & e \in \mathcal{A}_t^{\bar{p}}
\end{cases}.
\tag{16}
$$

**Lemma 9.** *Let $(\mathcal{X}, \mathcal{Y})$ be an $s - t$ cut of the edge-weighted directed graph $(\mathcal{U}, \mathcal{A}, w_\beta^{II})$, and $\mathcal{S} = \mathcal{X} \setminus \{s\}$. Then it holds that $\text{cost}(\mathcal{X}, \mathcal{Y}) = (2m - \lambda \cdot n_p) + \beta|\mathcal{S}| - p_2(\mathcal{S}, \lambda)$, where $p_2(\mathcal{S}, \lambda) = 2e(\mathcal{S}) - \lambda(|\mathcal{S}| + |\mathcal{S}_p| - 2|\mathcal{S} \cap \mathcal{S}_p|)$.*

*Proof.* Any edge from $\mathcal{X}$ to $\mathcal{Y}$ is contained in exactly one of the following sets: $\{(s,v) \in \mathcal{A}_s | v \in \mathcal{V} \setminus \mathcal{S}\}$, $\{(u,v) \in \mathcal{A}_\mathcal{E} | u \in \mathcal{S}, v \in \mathcal{V} \setminus \mathcal{S}\}$, $\{(v,t) \in \mathcal{A}_t^p | v \in \mathcal{S}\}$, $\{(v,t) \in \mathcal{A}_t^{\bar{p}} | v \in \mathcal{S}\}$. Therefore the cost of $(\mathcal{X}, \mathcal{Y})$ is

$$
\begin{aligned}
\text{cost}(\mathcal{X}, \mathcal{Y}) &= \sum_{(s,v) \in \mathcal{A}_s : v \in \mathcal{V} \setminus \mathcal{S}} w_\beta^{II}(s,v) + \sum_{(u,v) \in \mathcal{A}_\mathcal{E} : u \in \mathcal{S}, v \in \mathcal{V} \setminus \mathcal{S}} w_\beta^{II}(u,v) + \sum_{(u,t) \in \mathcal{A}_t^p : v \in \mathcal{S}} w_\beta^{II}(v,t) + \sum_{(u,t) \in \mathcal{A}_t^{\bar{p}} : v \in \mathcal{S}} w_\beta^{II}(v,t) \\
&= \sum_{v \in \mathcal{V} \setminus \mathcal{S}} \deg(v) + |\{(u,v) \in \mathcal{E} | u \in \mathcal{S}, v \in \mathcal{V} \setminus \mathcal{S}\}| + (\beta - \lambda)|\mathcal{S} \cap \mathcal{S}_p| + (\beta + \lambda)(|\mathcal{S}| - |\mathcal{S} \cap \mathcal{S}_p|) \\
&= 2m + \beta|\mathcal{S}| - 2e(\mathcal{S}) + \lambda|\mathcal{S}| - 2\lambda|\mathcal{S} \cap \mathcal{S}_p| = 2m + \beta|\mathcal{S}| - (p_2(\mathcal{S}, \lambda) + \lambda|\mathcal{S}_p|) \\
&= (2m - \lambda \cdot n_p) + \beta|\mathcal{S}| - p_2(\mathcal{S}, \lambda).
\end{aligned}
\tag{17}
$$

$\square$

Similarly to FADSG-I, let $(\mathcal{X}, \mathcal{Y})$ be the minimum $s - t$ cut (or max-flow) of $(\mathcal{U}, \mathcal{A}, w_\beta^{II})$. If $\mathcal{X} \neq \{s\}$, then $\mathcal{S} = \mathcal{X} \setminus \{s\}$ is a vertex subset that satisfies $\beta|\mathcal{S}| - p_2(\mathcal{S}, \lambda) \leq 0 \Leftrightarrow \frac{p_2(\mathcal{S}, \lambda)}{|\mathcal{S}|} \geq \beta$, and we can update the lower bound by $\beta$. If $\mathcal{X} = \{s\}$, then there is no $\mathcal{S} \subset \mathcal{V}$ that satisfies $\frac{p_2(\mathcal{S}, \lambda)}{|\mathcal{S}|} > \beta$, and the upper bound can be replaced by $\beta$. Therefore, we can again apply the same procedure as that of Lanciano et al. (2024, Algorithm 1), with a small change in the initial upper bound $\beta_{\text{ub}}^{(0)} \leftarrow (m - \lambda \cdot n_p)/2$, since $\lambda \cdot n_p$ is a constant.

### B.2 Upper bound on the number of binary search iterations for FADSG-I and II

Applying binary search to solve problems (2) and (4) requires specifying an initial interval $[L, U]$, where $(L, U)$ denote lower and upper bounds on the optimal value of (2) and (4), respectively, with $L < U$. The number of binary search iterations needed to solve problems (2) and (4) is $\log \frac{U-L}{\Delta}$, with $\Delta$ being the difference of any two distinct values of the objective function of FADSG-I and II. It is straight-forward to prove that $\Delta \geq O(1/n^2)$, similarly to Fazzone et al. (2022, Lemma 1).

#### B.2.1 For FADSG-I

For FADSG-I, we have the following bounds:

$$\min_{\mathbf{x} \in \{0,1\}^n} g(\lambda, \mathbf{x}) \geq \min_{\mathbf{x} \in \{0,1\}^n} \rho(\mathbf{x}) + \lambda \min_{\mathbf{x} \in \{0,1\}^n} r_1(\mathbf{x}) \Rightarrow L_1 = 0. \tag{18}$$

and

$$\max_{\mathbf{x} \in \{0,1\}^n} g(\lambda, \mathbf{x}) \leq \max_{\mathbf{x} \in \{0,1\}^n} \rho(\mathbf{x}) + \lambda \max_{\mathbf{x} \in \{0,1\}^n} r_1(\mathbf{x}) \Rightarrow U_1 = (n-1) + \lambda. \tag{19}$$

Thus, the number of iterations is

$$\log \frac{U_1 - L_1}{\Delta} \leq \log \frac{n-1+\lambda}{1/n^2} = \log(n-1+\lambda) + \log n^2 = O(\log(n+\lambda) + \log n). \tag{20}$$

#### B.2.2 For FADSG-II

For FADSG-II, we have the following bounds:

$$\min_{\mathbf{x} \in \{0,1\}^n} h(\lambda, \mathbf{x}) \geq \min_{\mathbf{x} \in \{0,1\}^n} \rho(\mathbf{x}) - \lambda \cdot \max_{\mathbf{x} \in \{0,1\}^n} r_2(\mathbf{x}) \Rightarrow L_2 = -\lambda \cdot n. \tag{21}$$

and

$$\max_{\mathbf{x} \in \{0,1\}^n} h(\lambda, \mathbf{x}) \leq \max_{\mathbf{x} \in \{0,1\}^n} \rho(\mathbf{x}) - \lambda \cdot \min_{\mathbf{x} \in \{0,1\}^n} r_2(\mathbf{x}) \Rightarrow U_2 = n - 1. \tag{22}$$

Thus, the number of iterations is

$$\log \frac{U_2 - L_2}{\Delta} \leq \log \frac{n-1+\lambda \cdot n}{1/n^2} = \log((1+\lambda) \cdot n - 1) + \log n^2 = O(\log(1+\lambda) + \log n). \tag{23}$$

## C   Identifying the end of the regularization path

In order to determine the choice of $\lambda$ that corresponds to the end of the regularization path of FADSG-I and II, we can consider analyzing the optimality conditions of problems (2) and (4). Our reasoning behind this step is that for *continuous, convex* regularization problems, studying the Karush-Kuhn-Tucker (KKT) conditions of the problem (which are necessary and sufficient for optimality) can facilitate the selection of the regularization parameter (see Bach et al. (2012)). However, while our formulations are tractable, they are *combinatorial* in nature, for which there is no analogue of the KKT conditions in general. We note that for the special case where the objective function being maximized is *supermodular*, the following characterization of the optimality conditions is known.

**Fact 1:** (Fujishige, 2005) *Consider the problem* $\max_{\mathcal{A} \subseteq \mathcal{V}} F(\mathcal{A})$, *where* $F(\mathcal{A})$ *is a supermodular function. Then, a set* $\mathcal{A}^*$ *is optimal if and only if*

$$F(\mathcal{A}^*) \geq F(\mathcal{B}), \forall \, \mathcal{B} \subseteq \mathcal{A}^* \text{ and } \forall \, \mathcal{B} \supseteq \mathcal{A}^*. \tag{24}$$

Hence, in order to certify that a candidate solution is optimal, it suffices to check only all its subsets and supersets, instead of examining all subsets of $\mathcal{V}$. The above property of supermodular functions can be viewed as a statement to the effect that local optimality implies global optimality, akin to convex functions in the continuous domain. Going forward, this property will prove useful.

### C.1 FADSG-I

Unfortunately, problem (2) is not supermodular, and hence (24) does not directly apply. To circumvent this limitation, we make the key observation that (2) *is equivalent to solving* a supermodular maximization problem of the following form.

**Fact 2:** *Let $v_\lambda^*$ be the optimal value of a given instance of (2). Then a maximizer of (2) is also a maximizer of the supermodular problem*

$$\max_{\mathcal{S} \subseteq \mathcal{V}} \left\{ F(\mathcal{S}) := p_1(\mathcal{S}, \lambda) - v_\lambda^* \cdot q(\mathcal{S}) \right\}. \tag{25}$$

*Furthermore, the optimal value of the above problem is $0$.*

Consequently, the optimality condition (24) can be applied for the above problem, which in turn will allow us to determine the smallest value of the regularization parameter $\lambda$ for which the solution of (2) is the densest protected subgraph of $\mathcal{S}_p$. Let $\mathcal{S}_p^* \subseteq \mathcal{S}_p$ denote the densest protected subset and $\rho(\mathcal{S}_p^*)$ denote its density. When the solution of (2) equals $\mathcal{S}_p^*$, the optimal value is $v_\lambda^* = \rho(\mathcal{S}_p^*) + \lambda$, since $r_1(\mathcal{S}_p^*) = 1$. We are now ready to prove Proposition 6 (restated below for convenience).

**Proposition 6.** *For $\lambda \geq \lambda_{\max} := [\max_{\mathcal{S} \supset \mathcal{S}_p} \{(\rho(\mathcal{S}) - \rho(\mathcal{S}_p^*))/(1 - \frac{n_p}{|\mathcal{S}|})\}]$, the solution of (2) is the densest protected subset $\mathcal{S}_p^*$.*

*Proof.* Applying the optimality conditions of (24) to problem (25), we obtain the condition

$$F(\mathcal{B}) \leq F(\mathcal{S}_p^*), \forall\, \mathcal{B} \supseteq \mathcal{S}_p^*. \tag{26}$$

Since $F(\mathcal{S}_p^*) = 0$, the condition simplifies to

$$\begin{aligned}
& p_1(\mathcal{B}, \lambda) - v_\lambda^* \cdot q(\mathcal{B}) \leq 0, \forall\, \mathcal{B} \supseteq \mathcal{S}_p^* \\
\Leftrightarrow\; & 2e(\mathcal{B}) + \lambda \cdot |\mathcal{B} \cap \mathcal{S}_p| - v_\lambda^* \cdot |\mathcal{B}| \leq 0, \\
\Leftrightarrow\; & \frac{2e(\mathcal{B})}{|\mathcal{B}|} + \lambda \cdot \frac{|\mathcal{B} \cap \mathcal{S}_p|}{|\mathcal{B}|} \leq \rho(\mathcal{S}_p^*) + \lambda.
\end{aligned} \tag{27}$$

Define $\rho(\mathcal{B}) := 2e(\mathcal{B})/|\mathcal{B}|$ to be the density of the subgraph induced by $\mathcal{B}$. Then, we obtain

$$\rho(\mathcal{B}) - \rho(\mathcal{S}_p^*) \leq \lambda \cdot \left[ 1 - \frac{|\mathcal{B} \cap \mathcal{S}_p|}{|\mathcal{B}|} \right], \forall\, \mathcal{B} \supseteq \mathcal{S}_p^*. \tag{28}$$

We now distinguish between two categories of supersets $\mathcal{B}$. Either we have **(i):** $\mathcal{S}_p \supseteq \mathcal{B} \supseteq \mathcal{S}_p^*$, or **(ii):** $\mathcal{B} \supset \mathcal{S}_p \supseteq \mathcal{S}_p^*$. If the first case is true, condition (28) reduces to

$$\rho(\mathcal{B}) \leq \rho(\mathcal{S}_p^*), \forall\, \mathcal{S}_p \supseteq \mathcal{B} \supseteq \mathcal{S}_p^*, \tag{29}$$

which is always guaranteed to hold (irrespective of $\lambda$) since $\mathcal{S}_p^*$ is the densest protected subset. This implies that we only need consider the second category of supersets. Then, (28) becomes

$$\lambda \geq \frac{\rho(\mathcal{B}) - \rho(\mathcal{S}_p^*)}{1 - \frac{|\mathcal{B} \cap \mathcal{S}_p|}{|\mathcal{B}|}}, \forall\, \mathcal{B} \supset \mathcal{S}_p. \tag{30}$$

Since $|\mathcal{B} \cap \mathcal{S}_p| = |\mathcal{S}_p| = n_p, \forall\, \mathcal{B} \supset \mathcal{S}_p$, we finally obtain

$$\lambda \geq \max_{\mathcal{B} \supset \mathcal{S}_p} \left\{ \frac{\rho(\mathcal{B}) - \rho(\mathcal{S}_p^*)}{1 - \frac{n_p}{|\mathcal{B}|}} \right\}. \tag{31}$$

Note that the optimality conditions (24) also require us to verify that the above choice of $\lambda_{\max}$ satisfies

$$F(\mathcal{B}) \leq F(\mathcal{S}_p), \forall\, \mathcal{B} \subseteq \mathcal{S}_p. \tag{32}$$

Note that for $\mathcal{B} = \emptyset$, the above condition is guaranteed to hold irrespective of $\lambda$. For subsets $\mathcal{B}$ satisfying $\emptyset \subset \mathcal{B} \subset \mathcal{S}_p$, we have $|\mathcal{B} \cap \mathcal{S}_p| = |\mathcal{B}|$. Consequently, (28) becomes

$$\rho(\mathcal{B}) - \rho(\mathcal{S}_p^*) \leq 0, \forall \, \emptyset \subset \mathcal{B} \subset \mathcal{S}_p, \tag{33}$$

which is again always satisfied irrespective of $\lambda$, since $\rho(\mathcal{S}_p^*)$ is the densest protected subset. The claim then follows. $\square$

## C.2 FADSG-II

We adopt the same line of reasoning as the previous subsection. Our starting point is the fact that given an instance of (4) with optimal value $v_\lambda^*$, a maximizer of (4) is also a maximizer of the following supermodular maximization problem

$$\max_{\mathcal{S} \subseteq \mathcal{V}} \left\{ F(\mathcal{S}) := p_2(\mathcal{S}, \lambda) - v_\lambda^* \cdot q(\mathcal{S}) \right\} \tag{34}$$

with optimal value equal to 0. This allows us to use the optimality conditions of the above problem (which are characterized by (24)) to determine the smallest value of $\lambda$ required so that the solution of (4) corresponds to the protected subset $\mathcal{S}_p$. Note that in such a case we have $v_\lambda^* = \rho(\mathcal{S}_p)$. We are now ready to prove Proposition 4.2.

**Proposition 10.** *Define the quantities*

$$\lambda_1 := \max_{\mathcal{B} \supset \mathcal{S}_p} \left\{ \frac{\rho(\mathcal{B}) - \rho(\mathcal{S}_p)}{1 - \frac{n_p}{|\mathcal{B}|}} \right\}, \lambda_2 := \max_{\emptyset \subset \mathcal{B} \subset \mathcal{S}_p} \left\{ \frac{\rho(\mathcal{B}) - \rho(\mathcal{S}_p)}{\frac{n_p}{|\mathcal{B}|} - 1} \right\}$$

*Then, for $\lambda \geq \max\{\lambda_1, \lambda_2\}$, the solution of* (4) *is the protected subset $\mathcal{S}_p$.*

*Proof.* Since $F(\cdot)$ is supermodular, the optimality conditions (24) assert that

$$F(\mathcal{B}) \leq F(\mathcal{S}_p), \forall \, \mathcal{B} \supseteq \mathcal{S}_p. \tag{35}$$

Since $F(\mathcal{S}_p) = 0$, the condition becomes

$$\begin{aligned}
&p_2(\mathcal{B}, \lambda) - v_\lambda^* \cdot q(\mathcal{B}) \leq 0, \forall \, \mathcal{B} \supseteq \mathcal{S}_p \\
\Leftrightarrow \, &\frac{2e(\mathcal{B})}{|\mathcal{B}|} - \lambda \cdot r_2(\mathcal{B}) \leq \rho(\mathcal{S}_p), \\
\Leftrightarrow \, &\rho(\mathcal{B}) - \rho(\mathcal{S}_p) \leq \lambda r_2(\mathcal{B})
\end{aligned} \tag{36}$$

For a superset $\mathcal{B} \supset \mathcal{S}_p$, we have $r_2(\mathcal{B}) = 1 - (n_p/|\mathcal{B}|)$. Thus we obtain the condition

$$\lambda \geq \lambda_1 := \max_{\mathcal{B} \supset \mathcal{S}_p} \left\{ \frac{\rho(\mathcal{B}) - \rho(\mathcal{S}_p)}{1 - \frac{n_p}{|\mathcal{B}|}} \right\}. \tag{37}$$

From (24), we also have the condition

$$F(\mathcal{B}) \leq F(\mathcal{S}_p), \forall \, \mathcal{B} \subseteq \mathcal{S}_p, \tag{38}$$

which can be simplified as

$$2e(\mathcal{B}) - \lambda \cdot [|\mathcal{B}| + |\mathcal{S}_p| - 2|\mathcal{B} \cap \mathcal{S}_p|] \leq \rho(\mathcal{S}_p) \cdot |\mathcal{B}|, \forall \, \mathcal{B} \subseteq \mathcal{S}_p. \tag{39}$$

A little calculation reveals that when $\mathcal{B} = \emptyset$, the above condition holds irrespective of $\lambda \geq 0$. Hence, we restrict our attention to subsets of the form $\emptyset \subset \mathcal{B} \subset \mathcal{S}_p$. In this case (39) becomes

$$\rho(\mathcal{B}) - \rho(\mathcal{S}_p) \leq \lambda r_2(\mathcal{B}), \forall \, \emptyset \subset \mathcal{B} \subset \mathcal{S}_p. \tag{40}$$

Since $r_2(\mathcal{B}) = (n_p/|\mathcal{B}|) - 1$, we finally obtain

$$\lambda \geq \lambda_2 := \max_{\emptyset \subset \mathcal{B} \subset \mathcal{S}_p} \left\{ \frac{\rho(\mathcal{B}) - \rho(\mathcal{S}_p)}{\frac{n_p}{|\mathcal{B}|} - 1} \right\}. \tag{41}$$

Combining the inequalities (37) and (41), we obtain

$$\lambda \geq \max\{\lambda_1, \lambda_2\}. \tag{42}$$

from which the claim follows. $\qquad\square$

## D  Analyzing the Price of Fairness

Recall that the lollipop graph on $n$ vertices (denoted as $\mathcal{L}_n$) comprises an unprotected clique of size $\sqrt{n}$ (denoted as $\mathcal{C}_{\sqrt{n}}$), while the protected vertices form a path graph on $n - \sqrt{n}$ vertices (denoted as $\mathcal{P}_{n-\sqrt{n}}$), with one end connected to $\mathcal{C}_{\sqrt{n}}$ (so that the overall graph is connected).

In order to analyze the price of fairness (PoF) on $\mathcal{L}_n$, the following facts will be useful. Let $\mathcal{C}_k$ denote a clique on $k$ vertices with density $\rho(\mathcal{C}_k) := k - 1$. Additionally, let $\mathcal{P}_k$ denote a path graph on $k$ vertices. A little calculation reveals that the density of the path graph is $\rho(\mathcal{P}_k) := 2 \cdot (1 - 1/k)$. Given two distinct vertex subsets $\mathcal{X}, \mathcal{Y}$ with no overlap, let $e(\mathcal{X}; \mathcal{Y})$ denote the number of edges with one endpoint in $\mathcal{X}$ and the other in $\mathcal{Y}$. Then, the number of edges induced by the combined vertex subset $\mathcal{X} \cup \mathcal{Y}$ can be expressed as

$$e(\mathcal{X} \cup \mathcal{Y}) = e(\mathcal{X}) + e(\mathcal{Y}) + e(\mathcal{X}; \mathcal{Y}). \tag{43}$$

**Proposition 11.** *For $n > 9$, the densest subgraph of $\mathcal{L}_n$ is the unprotected clique $\mathcal{C}_{\sqrt{n}}$.*

*Proof.* Consider a candidate subgraph comprising $r \leq \sqrt{n}$ unprotected vertices and $s \leq n - \sqrt{n}$ protected vertices. The solution can be expressed as $\mathcal{S} = \mathcal{C}_r \cup \mathcal{P}_s$, with density $\rho(\mathcal{S})$ given by

$$\rho(\mathcal{S}) = \frac{2e(\mathcal{C}_r \cup \mathcal{P}_s)}{r + s} = \frac{2[e(\mathcal{C}_r) + e(\mathcal{P}_s) + e(\mathcal{C}_r; \mathcal{P}_s)]}{r + s} \leq \frac{2[e(\mathcal{C}_r) + e(\mathcal{P}_s) + 1]}{r + s}$$
$$\leq \max\left\{ \frac{2e(\mathcal{C}_r)}{r}, \frac{2[e(\mathcal{P}_s) + 1]}{s} \right\} = \max\{\rho(\mathcal{C}_r), 2\} = \max\{r - 1, 2\}. \tag{44}$$

The first inequality stems from the fact that $e(\mathcal{C}_r; \mathcal{P}_s) \leq 1$, whereas the second inequality is due to the following fact: given positive numbers $a, b, c, d$, we have $(a + c)/(b + d) \leq \max\{a/b, c/d\}$. For $r > 3$, the maximum is attained by the first term (irrespective of $s$) - note that a necessary condition is $n > 9$. Hence we obtain

$$\rho(\mathcal{S}) \leq r - 1 \leq \sqrt{n} - 1, \tag{45}$$

for any candidate subgraph $\mathcal{S}$. Both inequalities are satisfied with equality when $\mathcal{S} = \mathcal{C}_r$ with $r = \sqrt{n}$, which establishes the desired claim. $\qquad\square$

We now turn towards formalizing this intuition. From Proposition 11, it follows that $\rho^*_{\mathcal{L}_n} = \sqrt{n} - 1$, and $l_1 = 0$. Hence, the admissible range of $\alpha$ is the unit interval $[0, 1]$ (which is the largest possible range). First, consider the case where the target fairness level $\alpha = 1$, which corresponds to extracting the subgraph $\mathcal{P}_{n-\sqrt{n}}$ induced by the protected vertices.[3] This case corresponds to the opposite extreme of DSG (where no fairness is imposed). It turns out that the PoF can be expressed as follows.

**Proposition 12.** $PoF(\mathcal{L}_n, 1) > 1 - [2/(\sqrt{n} - 1)]$.

*Proof.* For $n > 9$, have established that $\rho^*_{\mathcal{L}_n} = \sqrt{n} - 1$. Since $\rho(\mathcal{P}_{n-\sqrt{n}}) < 2$, it follows from the definition that $\text{PoF}(\mathcal{L}_n, 1) > 1 - [2/(\sqrt{n} - 1)]$. $\qquad\square$

---

[3]Note that by construction, the densest protected subset coincides with the entire protected subset in this case.

The above result is pessimistic as it demonstrates that in the worst-case, the price paid in extracting the fair component can be made arbitrarily close to 1, via a suitable choice of $n$. Additionally, it demonstrates that the lollipop graph is indeed a suitable candidate for analyzing the worst-case trade-off between density and fairness. Next, we present the proof of Proposition 7, in order to understand how the PoF is affected by altering the balance between the majority and minority vertices in the desired solution.

### D.1 Proof of Proposition 7

Recall that the desired proportion of unprotected and protected vertices in the extracted subgraph is $1 : \gamma$, where $\gamma \in \{1, 2, \cdots, n - \sqrt{n}\}$ is a positive integer. In this case, the target fairness level can be expressed as the ratio $\alpha_\gamma := \gamma/(1 + \gamma)$. We aim to show that

$$\text{PoF}(\mathcal{G}, \alpha_\gamma) \begin{cases} = \alpha_\gamma \left[ 1 - \frac{2}{\sqrt{n}-1} \right], & \alpha_\gamma \in \left\{ \frac{1}{2}, \cdots, 1 - \frac{1}{\sqrt{n}} \right\} \\ > 1 - \left[ \frac{1}{(\sqrt{n})} + \frac{2}{\sqrt{n}-1} \right], & \alpha_\gamma \in \left\{ 1 - \frac{1}{\sqrt{n}+1}, \cdots, 1 - \frac{1}{n-\sqrt{n}+1} \right\} \end{cases} \quad (46)$$

*Proof.* Consider the range $\gamma \in \{1, \cdots, \sqrt{n} - 1\}$, for which $\alpha_\gamma$ lies in the range $\{\frac{1}{2}, \cdots, 1 - \frac{1}{\sqrt{n}}\}$. Given a value $\gamma$ lying in the above range, the optimal fair densest subgraph comprises $\sqrt{n}$ unprotected vertices of $\mathcal{C}_{\sqrt{n}}$ and $\gamma\sqrt{n}$ protected vertices of the path graph $\mathcal{P}_{n-\sqrt{n}}$. The optimal density is

$$\rho^*_{\mathcal{L}_n}(\alpha_\gamma) = \frac{2e(\mathcal{C}_{\sqrt{n}} \cup \mathcal{P}_{\gamma\sqrt{n}})}{(1+\gamma)\sqrt{n}} = \frac{(\sqrt{n} - 1 + 2\gamma)}{(1+\gamma)} \quad (47)$$

Hence, the PoF is

$$\text{PoF}(\mathcal{L}_n, \alpha_\gamma) = 1 - \frac{\rho^*_{\mathcal{L}_n}(\alpha_\gamma)}{\rho^*_{\mathcal{L}_n}} = \frac{\gamma}{1+\gamma} \left[ 1 - \frac{2}{\sqrt{n}-1} \right]. \quad (48)$$

Next, we consider $\gamma$ in the range $\{\sqrt{n}, \cdots, n - \sqrt{n}\}$ for which $\alpha_\gamma$ ranges from $\{1 - \frac{1}{\sqrt{n}+1}, \cdots, 1 - \frac{1}{n-\sqrt{n}+1}\}$. In this range, the optimal fair densest subgraph comprises the $n - \sqrt{n}$ vertices of the protected subgraph $\mathcal{P}_{n-\sqrt{n}}$ and $r < \sqrt{n}$ unprotected vertices from $\mathcal{C}_{\sqrt{n}}$ such that $r$ satisfies $r = (n - \sqrt{n})/\gamma$. The density of the solution is given by

$$\rho^*_{\mathcal{L}_n}(\alpha_\gamma) = \frac{2e(\mathcal{C}_r \cup \mathcal{P}_{n-\sqrt{n}})}{(1+\gamma)r} = \frac{(r - 1 + 2\gamma)}{(1+\gamma)} = \frac{(n - \sqrt{n} - \gamma + 2\gamma^2)}{\gamma(1+\gamma)} = \frac{\sqrt{n}(\sqrt{n}-1)}{\gamma(1+\gamma)} + \frac{2\gamma - 1}{(1+\gamma)} \quad (49)$$

Hence, the PoF in this case can be expressed as

$$\begin{aligned} \text{PoF}(\mathcal{L}_n, \alpha_\gamma) &= 1 - \left[ \frac{\sqrt{n}}{\gamma(1+\gamma)} + \frac{2\gamma-1}{(1+\gamma)(\sqrt{n}-1)} \right] > 1 - \left[ \frac{\sqrt{n}}{\gamma^2} + \frac{2}{\sqrt{n}-1} \right] \geq 1 - \left[ \frac{\sqrt{n}}{(\sqrt{n})^2} + \frac{2}{\sqrt{n}-1} \right] \\ &= 1 - O(1/\sqrt{n}) \end{aligned} \quad (50)$$

$\square$

### D.2 Equivalence of $\delta$ and $\alpha$

As we have already explained in Section 5, the solution of DSG without any fairness considerations is the clique $\mathcal{C}_{\sqrt{n}}$ that contains the unprotected vertices. For this solution, it holds that $r_2(\mathcal{S})$ attains its maximum value, which is $r_2(\mathcal{S}) = \sqrt{n}$. As we decrease the target fairness value $\delta$ of FADSG-II from this upper bound $\delta_u = \sqrt{n}$, the solution will gradually include more and more protected vertices from the protected subgraph $\mathcal{P}_{n-\sqrt{n}}$, until we reach the point that we have the whole graph $\mathcal{G}$, for which point it holds that $r_2(\mathcal{S}) = \frac{1}{\sqrt{n}}$. Up to this point, it always holds that we have a certain fraction (say $1/x < 1$) of the protected vertices in the solution, i.e., $\frac{|\mathcal{S} \cap \mathcal{S}_p|}{|\mathcal{S}_p|} = \frac{1}{x} < 1 \Leftrightarrow |\mathcal{S} \cap \mathcal{S}_p| = \frac{n_p}{x} = \frac{n-\sqrt{n}}{x}$. This implies that

$$r_1(\mathcal{S}) = \frac{|\mathcal{S} \cap \mathcal{S}_p|}{|\mathcal{S}|} = \frac{\frac{n-\sqrt{n}}{x}}{\sqrt{n} + \frac{n-\sqrt{n}}{x}} \Leftrightarrow r_1(\mathcal{S}) = \frac{\sqrt{n}-1}{\sqrt{n} + (x-1)}, \quad (51)$$

Moreover, it follows that

$$
\begin{aligned}
r_2(\mathcal{S}) = 1 + \frac{n_p}{|\mathcal{S}|} - 2r_1(\mathcal{S}) &= 1 + \frac{n_p - 2|\mathcal{S} \cap \mathcal{S}_p|}{|\mathcal{S}|} = 1 + \frac{n_p - 2\frac{1}{x}n_p}{|\mathcal{S}|} = 1 + \frac{\frac{1}{x}(x-2)n_p}{|\mathcal{S}|} \\
&= \frac{\sqrt{n} + \frac{x-1}{x}n_p}{\sqrt{n} + \frac{1}{x}n_p} = \frac{\sqrt{n} + \frac{x-1}{x}(n - \sqrt{n})}{\sqrt{n} + \frac{1}{x}(n - \sqrt{n})} = \frac{1 + \frac{x-1}{x}(\sqrt{n} - 1)}{1 + \frac{1}{x}(\sqrt{n} - 1)} = \frac{x + (x-1)\sqrt{n} - x + 1}{x + \sqrt{n} - 1}
\end{aligned}
$$

$$
\Leftrightarrow r_2(\mathcal{S}) = \frac{(x-1)\sqrt{n} + 1}{\sqrt{n} + (x-1)}
$$

$$
\Leftrightarrow x = \frac{(1 + r_2(\mathcal{S}))(1 - \sqrt{n})}{r_2(\mathcal{S}) - \sqrt{n}}, \tag{52}
$$

which implies that

$$
r_1(\mathcal{S}) = \frac{(n + r_2(\mathcal{S})) - (r_2(\mathcal{S}) + 1)\sqrt{n}}{n - 1}. \tag{53}
$$

If we further decrease the target fairness value $\delta$, the solution will contain all of the protected vertices and it is going to start removing the unprotected ones. Hence, it will always hold that $\frac{|\mathcal{S} \cap \mathcal{S}_p|}{|\mathcal{S}_p|} = 1 \Leftrightarrow |\mathcal{S} \cap \mathcal{S}_p| = n_p$. This results in

$$
r_2(\mathcal{S}) = 1 + \frac{n_p}{|\mathcal{S}|} - 2\frac{|\mathcal{S} \cap \mathcal{S}_p|}{|\mathcal{S}|} \Leftrightarrow |\mathcal{S}| = \frac{n_p - 2|\mathcal{S} \cap \mathcal{S}_p|}{\delta - 1} = \frac{n_p}{1 - r_2(\mathcal{S})} \tag{54}
$$

and

$$
r_1(\mathcal{S}) = \frac{|\mathcal{S} \cap \mathcal{S}_p|}{|\mathcal{S}|} = \frac{n_p}{\frac{n_p}{1 - r_2(\mathcal{S})}} \Leftrightarrow r_1(\mathcal{S}) = 1 - r_2(\mathcal{S}). \tag{55}
$$

Finally, we get the following mapping of $\delta$ to $\alpha$:

- if $\delta > \frac{1}{\sqrt{n}}$, with $\frac{|\mathcal{S} \cap \mathcal{S}_p|}{n_p} = \frac{1}{x} < 1$, then

  $\alpha = \frac{\sqrt{n} - 1}{\sqrt{n} + (x - 1)} = \frac{(n + \delta) - (\delta + 1)\sqrt{n}}{n - 1}$.      [corresponds to the 1st branch of Eq. (46)]

- if $\delta \leq \frac{1}{\sqrt{n}}$, with $\frac{|\mathcal{S} \cap \mathcal{S}_p|}{n_p} = 1 \Leftrightarrow |\mathcal{S} \cap \mathcal{S}_p| = n_p$ , then

  $\alpha = 1 - \delta$.      [corresponds to the 2nd branch of Eq. (46)]

## E  Super-Greedy++

Super-Greedy++ is a efficient multi-stage greedy algorithm designed to retrieve high-quality solutions for problems referred to as Densest Supermodular Subset (DSS), i.e., maximization of $F(\mathcal{S})/|\mathcal{S}|$, for $F : 2^{\mathcal{V}} \to \mathbb{R}$. This algorithm can extract $(1 - \epsilon)$-approximate solutions, for $\epsilon \in (0, 1)$ under some conditions in $O(1/\epsilon^2)$ iterations, as the following result of Chekuri et al. (2022) indicates.

**Theorem 13.** *Let $F : 2^{\mathcal{V}} \to \mathbb{R}_{\geq 0}$ be a normalized, non-negative, and monotone supermodular function over the ground set $\mathcal{V}$, with $n = |\mathcal{V}|$. Let $\epsilon \in (0, 1)$. Let $\Delta = \max_{v \in \mathcal{V}} F(v|\mathcal{V})$ and $OPT$ the maximum of $F(\mathcal{S})/|\mathcal{S}|$ over all $\mathcal{S} \subseteq \mathcal{V}$. For $T \geq O\left(\frac{\Delta \cdot \ln n}{OPT \cdot \epsilon^2}\right)$, Super-Greedy++ outputs a $(1 - \epsilon)$-approximate solution to DSS.*

Pseudo-code of the algorithm is provided in (Algorithm 1). In order to implement this algorithm for FADSG-I and II, one has to derive the expressions of $F(v|\mathcal{S} - v)$, which are provided in Section E.1. For programming simplicity, our implementation at this point does not include heaps. Instead, we adopt a direct approach in which each time we recompute the values of $F(v|\mathcal{S} - v)$ and extract the minimum, while updating vertex degrees after peeling. Therefore, the computational complexity per iteration of our implementation is $O(n^2)$, resulting in an overall complexity of $O(n^2/\epsilon^2)$.

### E.1  Analytical Expressions of $F(v\,|\,\mathcal{S} - v)$.

As defined in Section 1, $f(v|\mathcal{S}) := f(\mathcal{S} \cup \{v\}) - f(\mathcal{S})$. Thus, $F(v\,|\,\mathcal{S} - v) = F(\mathcal{S}) - F(\mathcal{S} - v)$, where $\{\mathcal{S} - v\}$ is the set $\mathcal{S}$ without the vertex $v$. Furthermore, with $\deg_{\mathcal{S}}(v)$, we denote the degree of $v$ in the induced set $\mathcal{S}$.

---

**Algorithm 1** SUPER-GREEDY++

---

> **Input:** $F : 2^{\mathcal{V}} \to \mathbb{R}_{\geq 0}$, $T \in \mathbb{Z}_+$
> **Output:** $\mathcal{S}_{\text{densest}}$
> $\mathcal{S}_{\text{densest}} \leftarrow \mathcal{V}$
> for all $v \in \mathcal{V}$, set $\ell_v^{(0)} = 0$
> **for** $i : 1 \to T$ **do**
>     $\mathcal{S}_{i,1} \leftarrow \mathcal{V}$
>     **for** $j : 1 \to n$ **do**
>         $v^* \in \text{argmin}_{v^* \in \mathcal{S}_{i,j}} \, \ell_v^{(i-1)} + F(v \,|\, \mathcal{S}_{i,j} - v)$
>         $\ell_{v^*}^{(i)} \leftarrow \ell_{v^*}^{(i-1)} + F(v^* \,|\, \mathcal{S}_{i,j} - v^*)$
>         $\mathcal{S}_{i,j+1} \leftarrow \mathcal{S}_{i,j} - v^*$
>         **if** $\frac{F(\mathcal{S}_{i,j})}{|\mathcal{S}_{i,j}|} > \frac{F(\mathcal{S}_{\text{densest}})}{|\mathcal{S}_{\text{densest}}|}$ **then**
>             $\mathcal{S}_{\text{densest}} \leftarrow \mathcal{S}_{i,j}$
>         **end if**
>     **end for**
> **end for**

---

**FADSG-I:** In FADSG-I, the objective function is $g(\mathbf{x}, \lambda) = (\mathbf{x}^\top \mathbf{A}\mathbf{x} + \lambda \cdot \mathbf{e}_p^\top \mathbf{x})/(\mathbf{e}^\top \mathbf{x})$. For a given $\lambda$, we can set $F(\mathbf{x}) = \mathbf{x}^\top \mathbf{A}\mathbf{x} + \lambda \cdot \mathbf{e}_p^\top \mathbf{x} = p_1(\mathbf{x}, \lambda)$ and express it in terms of $\mathcal{S} \subseteq \mathcal{V}$, to get $F(\mathcal{S}) = 2e(\mathcal{S}) + \lambda|\mathcal{S} \cap \mathcal{S}_p|$. We also have that:

$$F(\mathcal{S} - v) = \begin{cases} 2e(\mathcal{S}) - 2\deg_{\mathcal{S}}(v) + \lambda(|\mathcal{S} \cap \mathcal{S}_p| - 1), & \text{if } v \in \mathcal{S} \cap \mathcal{S}_p \\ 2e(\mathcal{S}) - 2\deg_{\mathcal{S}}(v) + \lambda|\mathcal{S} \cap \mathcal{S}_p|, & \text{if } v \notin \mathcal{S} \cap \mathcal{S}_p \end{cases}. \tag{56}$$

Thus:

$$F(v \,|\, \mathcal{S} - v) = F(\mathcal{S}) - F(\mathcal{S} - v) = \begin{cases} 2\deg_{\mathcal{S}}(v) + \lambda, & \text{if } v \in \mathcal{S} \cap \mathcal{S}_p \\ 2\deg_{\mathcal{S}}(v), & \text{if } v \notin \mathcal{S} \cap \mathcal{S}_p \end{cases}. \tag{57}$$

**FADSG-II:** In FADSG-II, the objective function is $h(\mathbf{x}, \lambda) = (\mathbf{x}^\top \mathbf{A}\mathbf{x} - \lambda\|\mathbf{x} - \mathbf{e}_p\|_2^2)(\mathbf{e}^\top \mathbf{x})$. If we expand the norm of the numerator we get

$$\|\mathbf{x} - \mathbf{e}_p\|_2^2 = \|\mathbf{x}\|_2^2 + \|\mathbf{e}_p\|_2^2 - 2\mathbf{e}_p^\top \mathbf{x} = |\mathcal{S}| + |\mathcal{S}_p| - 2|\mathcal{S} \cap \mathcal{S}_p|. \tag{58}$$

For a given $\lambda$, we can set $F(\mathbf{x}) = \mathbf{x}^\top \mathbf{A}\mathbf{x} - \lambda\|\mathbf{x} - \mathbf{e}_p\|_2^2$ and express it in terms of $\mathcal{S} \subseteq \mathcal{V}$, to get $F(\mathcal{S}) = 2e(\mathcal{S}) + \lambda(|\mathcal{S}| + |\mathcal{S}_p| - 2|\mathcal{S} \cap \mathcal{S}_p|)$. We also have that:

$$\begin{aligned} F(S - v) &= \begin{cases} 2e(\mathcal{S}) - 2\deg_{\mathcal{S}}(v) - \lambda(|\mathcal{S}| - 1 + |\mathcal{S}_p| - 2(|\mathcal{S} \cap \mathcal{S}_p| - 1)), & \text{if } v \in \mathcal{S} \cap \mathcal{S}_p \\ 2e(\mathcal{S}) - 2\deg_{\mathcal{S}}(v) - \lambda(|\mathcal{S}| - 1 + |\mathcal{S}_p| - 2|\mathcal{S} \cap \mathcal{S}_p|), & \text{if } v \notin \mathcal{S} \cap \mathcal{S}_p \end{cases} \\ &= \begin{cases} 2e(\mathcal{S}) - 2\deg_{\mathcal{S}}(v) - \lambda(|\mathcal{S}| + |\mathcal{S}_p| - 2|\mathcal{S} \cap \mathcal{S}_p|) - \lambda, & \text{if } v \in \mathcal{S} \cap \mathcal{S}_p \\ 2e(\mathcal{S}) - 2\deg_{\mathcal{S}}(v) - \lambda(|\mathcal{S}| + |\mathcal{S}_p| - 2|\mathcal{S} \cap \mathcal{S}_p|) + \lambda, & \text{if } v \notin \mathcal{S} \cap \mathcal{S}_p \end{cases}. \end{aligned} \tag{59}$$

Thus:

$$F(v \,|\, \mathcal{S} - v) = F(\mathcal{S}) - F(\mathcal{S} - v) = \begin{cases} 2\deg_{\mathcal{S}}(v) + \lambda, & \text{if } v \in \mathcal{S} \cap \mathcal{S}_p \\ 2\deg_{\mathcal{S}}(v) - \lambda, & \text{if } v \notin \mathcal{S} \cap \mathcal{S}_p \end{cases}. \tag{60}$$

## F  Dataset details.

A summary of the statistics of all the datasets used in our experiments can be found in Table 2.

**Political Books Dataset (PolBooks):** The vertices of the network are books on US politics included in the Amazon catalog, with an edge connecting two books if they are frequently co-purchased by the same buyers[4]. Each book is categorized based on its political stance, with possible labels including liberal, neutral, and conservative. In our experiments, we focused solely on the subgraph formed by liberal and conservative

---

[4]https://websites.umich.edu/~mejn/netdata/

Table 2: Full summary of dataset statistics: number of vertices ($n$), number of edges ($m$), and size of protected subset ($n_p$).

| DATASET | $n$ | $m$ | $n_p$ |
|---|---|---|---|
| POLBOOKS | 92 | 374 | 43 |
| POLBLOGS | 1222 | 16717 | 586 |
| AMAZON (10) | $3699 \pm 2764$ | $22859 \pm 18052$ | $1927 \pm 2537$ |
| DEEZER | 28281 | 92752 | 12538 |
| LASTFM (19) | 7624 | 27806 | $424 \pm 454$ |
| GITHUB | 37700 | 289003 | 9739 |
| TWITCH (6) | $5686 \pm 2393$ | $71519 \pm 45827$ | $2523 \pm 1787$ |
| TWITTER | 18470 | 48053 | 11355 |

books, resulting in 92 vertices (43 associated with a conservative and 49 with a liberal worldview) connected by a total of 374 edges, as curated in Anagnostopoulos et al. (2020; 2024).

**Political Blogs Dataset (PolBlogs)** (Adamic & Glance, 2005): The vertices of the network are weblogs on US politics, with edges representing hyperlinks. Each blog is categorized by its political stance, left or right, with the left ones composing the protected set.

**Amazon Products Metadata (Amazon)** (Ni et al., 2019): The vertices represent Amazon products, and the edges denote frequent co-purchasing product pairs. The attribute associated with each product is their category. In our experiments we used 10 different such datasets (see Table 3), as assembled in Anagnostopoulos et al. (2020; 2024). We name them based on the initials of the categories.

Table 3: Summary of Amazon Product Metadata and Twitch Users Social Networks datasets statistics: the number of vertices ($n$), the number of edges ($m$), and size of protected subset ($n_p$).

| DATASET | $n$ | $m$ | $n_p$ |
|---|---|---|---|
| AMAZON | $3699 \pm 2764$ | $22859 \pm 18052$ | $1927 \pm 2537$ |
| AMAZON B | 230 | 592 | 83 |
| AMAZON PS | 1809 | 14099 | 230 |
| AMAZON IS | 2009 | 8341 | 1495 |
| AMAZON HPC | 3011 | 10004 | 1352 |
| AMAZON OP | 2281 | 16542 | 462 |
| AMAZON AH | 10378 | 53679 | 8163 |
| AMAZON ACS | 5056 | 33827 | 987 |
| AMAZON G | 6435 | 56553 | 5376 |
| AMAZON SO | 3216 | 19331 | 604 |
| AMAZON TMI | 2565 | 15619 | 520 |
| TWITCH | $5686 \pm 2393$ | $71519 \pm 45827$ | $2523 \pm 1787$ |
| TWITCH DE | 9498 | 153138 | 5742 |
| TWITCH ENGB | 7126 | 35324 | 3888 |
| TWITCH ES | 4648 | 59382 | 1360 |
| TWITCH FR | 6549 | 112666 | 2415 |
| TWITCH PTBR | 1912 | 31299 | 661 |
| TWITCH RU | 4385 | 37304 | 1075 |

**LastFM Asia Social Network (LastFM)** (Rozemberczki & Sarkar, 2020): The vertices represent users of LastFM from Asian countries, and the edges are mutual follower connections among them. The attribute associated with each user is their location.

For the LastFM Asia Social Network dataset, we choose each time a different group to be the protected one, and we denote the resulting dataset as LASTFM-I, with I being the index of the set that is considered

protected. The number of vertices of each protected set can be found in Table 4. We also consider the LASTFM-100 network, for which we combine the vertices of all locations with less than 100 vertices and consider them as the protected set.

Table 4: Number of protected vertices $n_p$ for each of the LASTFM-I datasets.

| I | 0 | 1 | 2 | 3 | 4 | 5 | 6 | 7 | 8 | 9 | 10 | 11 | 12 | 13 | 14 | 15 | 16 | 17 | 100 |
|---|---|---|---|---|---|---|---|---|---|---|----|----|----|----|----|----|----|----|-----|
| $n_p$ | 1098 | 54 | 73 | 515 | 16 | 391 | 655 | 82 | 468 | 58 | 1303 | 138 | 57 | 63 | 570 | 257 | 254 | 1572 | 387 |

**Deezer Europe Social Network (Deezer)** (Rozemberczki & Sarkar, 2020): The vertices represent users of Deezer from European countries, and the edges are mutual follower connections among them. The attribute associated with each user is their gender.

**GitHub Developers (GitHub)** (Rozemberczki et al., 2019): The vertices represent developers in GitHub who have starred at least 10 repositories, and the edges are mutual follower relationships between them. The attribute associated with each user is whether they are web or a machine learning developers.

**Twitch Users Social Networks (Twitch)** (Rozemberczki et al., 2019): The vertices represent users of Twitch that stream in a certain language, and the edges are mutual friendships between them. The attribute associated with each user is if the user is using mature language.

**Twitter Retweet Political Network (Twitter)** (Rossi & Ahmed, 2015; Conover et al., 2011): The vertices are Twitter users, and the edges represent whether the users have retweeted each other. The attribute associated with each user is their political orientation (Tsioutsiouliklis et al., 2021).

## G  Details of Experiments

In order to find the appropriate $\lambda$ value for the desired target fairness level that we had for each experiment, we performed bisection on $\lambda$ values. The lower bound was always 0 and the upper was $\lambda_{\max}$. The value of $\lambda_{\max}$ was computed experimentally for each dataset. We specifically chose the value of $\lambda_{\max}$ based on our observation that it led to the densest protected subgraph for FADSG-I and the entire protected subgraph for FADSG-II. Moreover, increasing $\lambda_{\max}$ beyond this point did not alter the desired outcomes. The specific values employed for each dataset and formulation are detailed in Table 6. Furthermore, this bisection was performed with a tolerance on $\lambda$, $\varepsilon = 10^{-9}$. Thus, the number of iterations for each bisection experiment was bounded by $\left\lceil \log_2 \frac{\lambda_{\max}}{\varepsilon} \right\rceil$. Furthermore, in recent work of Fazzone et al. (2022), a stopping criterion for the number of iterations in the SUPER-GREEDY++ algorithm is being introduced, ensuring a certain level of solution's quality. However, we observed that this criterion tends to be conservative for our purposes. Through experimentation with various datasets, we determined that a value of $T = 5$ iterations provided satisfactory results across all our datasets in our implementation of SUPER-GREEDY++. Table 5 shows an example of the running time of each algorithm, on AMAZON TMI dataset.

All experiments were performed on a single machine, with Intel i7-13700K CPU @ 3.4GHz and 128GB of main memory running Python 3.12.0. The code of the baselines were obtained via personal communication with the respective authors.

Table 5: Comparison of running time of each algorithm for AMAZON TMI dataset. Using heaps will drastically reduce the complexity of our algorithm by $O(n)$.

|  | DDSP | PS | FPS | 2-DFSG | FADSG-I (OURS) |
|---|---|---|---|---|---|
| TIME (SEC) | 0.135372 | 0.371171 | 0.167832 | 10.079753 | 918 |

Table 6: $\lambda_{\max}$ values for each dataset and formulation.

| DATASET | FADSG-I | FADSG-II | DATASET | FADSG-I | FADSG-II |
|---|---|---|---|---|---|
| AMAZON B | 20 | 12 | POLBOOKS | 1.2 | 5 |
| AMAZON PS | 18 | 50 | POLBLOGS | 100 | 100 |
| AMAZON IS | 35 | 150 | DEEZER | 40 | 125 |
| AMAZON HPC | 35 | 70 | LASTFM *(ALL)* | 40 | 50 |
| AMAZON OP | 50 | 50 | GITHUB | 100 | 200 |
| AMAZON AH | 40 | 200 | TWITCH *(ALL)* | 100 | 200 |
| AMAZON ACS | 40 | 50 | TWITTER | 150 | 200 |
| AMAZON G | 40 | 140 | | | |
| AMAZON SO | 40 | 50 | | | |
| AMAZON TMI | 40 | 50 | | | |

# H    Additional Experimental Results

In this section, we offer further insights by presenting additional results obtained from various datasets, aiming to reinforce the validity of our findings.

## H.1    FADSG-I:

In Figure 11 a comparison with prior art is illustrated, for the remaining datasets. Once more, we exclusively showcase datasets where the DSG solution exhibits a fairness level lower than the desired one of 0.5.

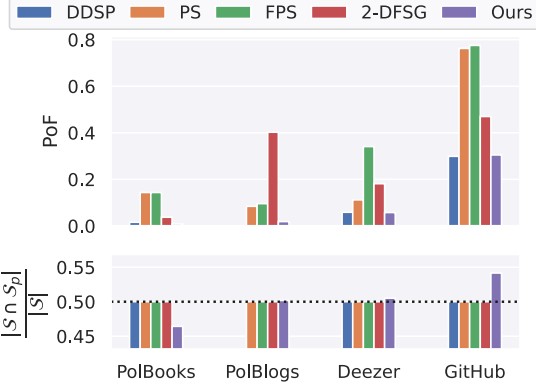

Figure 11: Comparing induced perfectly balanced fair subsets of prior art and FADSG-I (Ours), on POLBOOKS, BLOGS, DEEZER and GITHUB datasets. Top: PoF. Bottom: Fraction of protected vertices in induced subsets.

Figures 12, 13 and 14 illustrate the flexibility of our definition of fairness level $\alpha$. They reveal that certain desired fairness levels may be unattainable. As explained in Section 7.1, this limitation arises from the inherent structure of the graph, which does not naturally accommodate certain fairness levels. This can be clearly observed in Figure 13, where the right subfigure exhibits a notable discontinuity just before the curve reaches the value of 0.6, jumping to a value greater than 0.8. This is translated in the left subfigure, to an attained fairness level is slightly greater than 0.8, despite the desired level being 0.6.

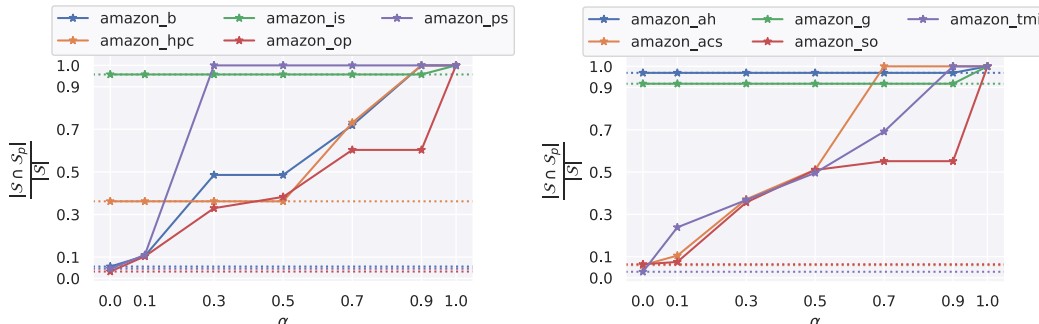

Figure 12: Fraction of protected vertices in each induced subgraph of FADSG-I, for various values of target fairness level $\alpha$, for the AMAZON datasets. The dotted lines represent the fairness value in the solution of classic DSG, which we want to enhance.

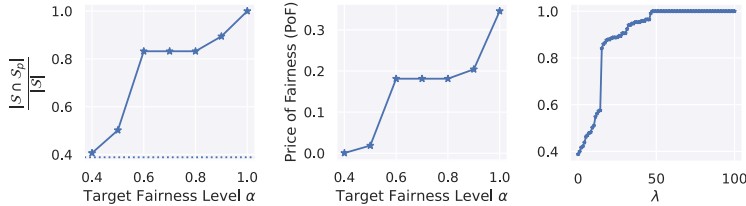

Figure 13: POLBLOGS dataset - FADSG-I. Left: Fraction of protected vertices in each induced subgraph, for various values of target fairness level $\alpha$. Middle: PoF for various values of target fairness level $\alpha$. The dotted lines represent the fairness value in the solution of DSG. Right: Fraction of protected vertices in each induced subgraph, for various values of the regularization parameter $\lambda$.

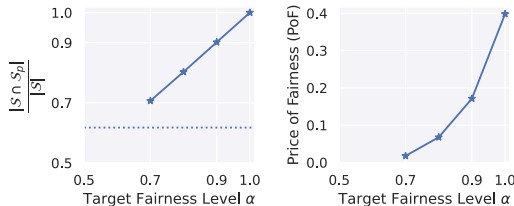

Figure 14: TWITTER dataset - FADSG-I. Left: Fraction of protected vertices in each induced subgraph, for various values of target fairness level $\alpha$. The dotted lines represent the fairness value in the solution of DSG. Right: PoF for various values of target fairness level $\alpha$.

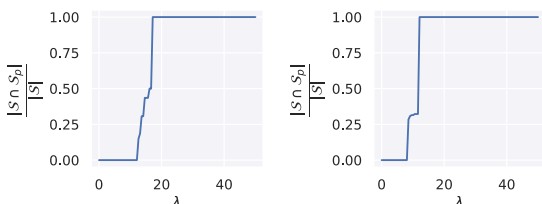

Figure 15: Fraction of protected vertices in the induced subset, as function of $\lambda$, for FADSG-I. (Left: LASTFM-12 - Right: LASTFM-13).

## H.2 FADSG-II:

In Figures 16, 17 and 18, we compare FADSG-II with DSG. The top sections show what fraction of all of the protected nodes belongs in the solutions, and the bottom ones depict the PoF for FADSG-II. Again, for some datasets the solution is not exactly the desired one, which is something that can be attributed to the structure of each graph, as it can be seen in Figure 19.

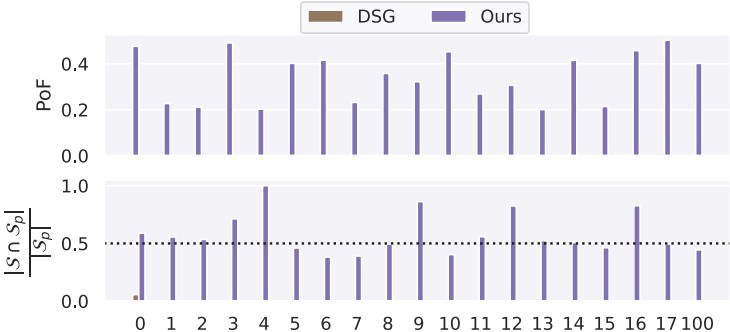

Figure 16: LASTFM datasets. Bottom: PoF of FADSG-II for $\delta = 1$. Top: Comparing FADSG-II (Ours) for $\delta = 1$ with DSG. Fraction of the protected subset that belongs in induced subsets.

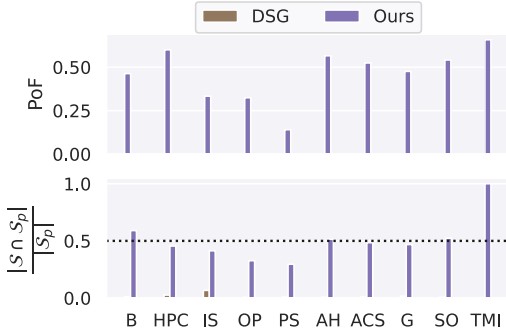

Figure 17: AMAZON datasets. Bottom: PoF of FADSG-II for $\delta = 1$. Top: Comparing FADSG-II (Ours) for $\delta = 1$ with DSG. Fraction of the protected subset that belongs in induced subsets.

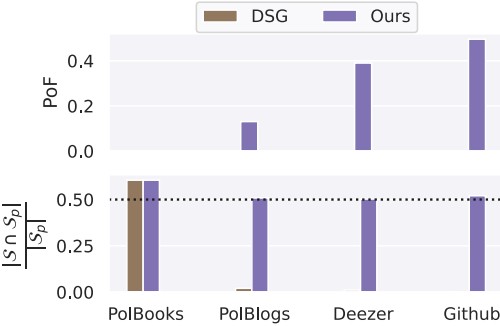

Figure 18: Rest of the datasets. Bottom: PoF of FADSG-II for $\delta = 1$. Top: Comparing FADSG-II (Ours) for $\delta = 1$ with DSG. Fraction of the protected subset that belongs in induced subsets.

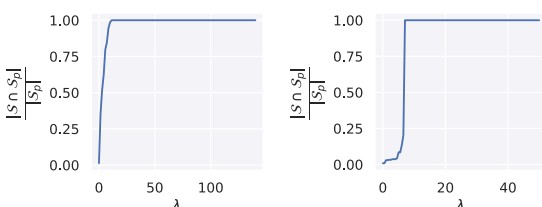

Figure 19: The fraction of all of the protected vertices included in induced subsets as function of $\lambda$, for FADSG-II. (Left: AMAZON G - Right: AMAZON OP).

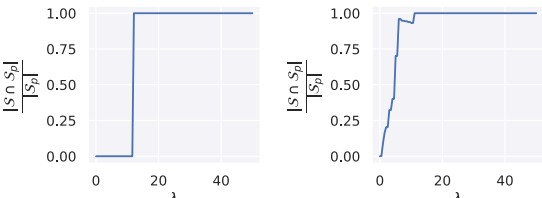

Figure 20: Fraction of all of the protected vertices that belong in the induced subset, as function of $\lambda$, for FADSG-II. (Left: LASTFM-4 - Right: LASTFM-14).

### H.3 Resulting proportions of protected vertices in FADSG-I solution

Table 7: Resulting proportions of protected vertices in FADSG-I solution $\mathcal{S}$. $n_p$ the total number of protected vertices, $|\mathcal{S} \cap \mathcal{S}_p|$ the number of protected vertices in $\mathcal{S}$, $\frac{|\mathcal{S}\cap\mathcal{S}_p|}{|\mathcal{S}|}$ the proportion of protected vertices in $\mathcal{S}$, and $\frac{|\mathcal{S}\cap\mathcal{S}_p|}{|\mathcal{S}_p|}$ the proportion among all of the protected vertices that belong in $\mathcal{S}$.

| | | | | | LASTFM DATASET | | | | |
| DATASET | $n_p = |\mathcal{S}_p|$ | $|\mathcal{S} \cap \mathcal{S}_p|$ | $\frac{|\mathcal{S}\cap\mathcal{S}_p|}{|\mathcal{S}|}$ | $\frac{|\mathcal{S}\cap\mathcal{S}_p|}{|\mathcal{S}_p|}$ | PROTECTED SET | $n_p = |\mathcal{S}_p|$ | $|\mathcal{S} \cap \mathcal{S}_p|$ | $\frac{|\mathcal{S}\cap\mathcal{S}_p|}{|\mathcal{S}|}$ | $\frac{|\mathcal{S}\cap\mathcal{S}_p|}{|\mathcal{S}_p|}$ |
|---|---|---|---|---|---|---|---|---|---|
| POLBOOKS | 42 | 26 | 1.0 | 0.6046 | 0 | 1098 | 62 | 1.0 | 0.0565 |
| POLBLOGS | 586 | 117 | 1.0 | 0.1997 | 1 | 54 | 10 | 1.0 | 0.1852 |
| AMAZON B | 83 | 10 | 1.0 | 0.1205 | 2 | 73 | 26 | 1.0 | 0.3562 |
| AMAZON PS | 230 | 17 | 1.0 | 0.0739 | 3 | 515 | 43 | 1.0 | 0.0835 |
| AMAZON IS | 1495 | 134 | 1.0 | 0.0896 | 4 | 16 | 16 | 1.0 | 1.0000 |
| AMAZON HPC | 1352 | 25 | 1.0 | 0.0185 | 5 | 391 | 44 | 1.0 | 0.1125 |
| AMAZON OP | 462 | 8 | 1.0 | 0.0173 | 6 | 655 | 46 | 1.0 | 0.0702 |
| AMAZON AH | 8163 | 62 | 1.0 | 0.0076 | 7 | 82 | 14 | 1.0 | 0.1707 |
| AMAZON ACS | 987 | 35 | 1.0 | 0.0355 | 8 | 468 | 35 | 1.0 | 0.0748 |
| AMAZON G | 5376 | 65 | 1.0 | 0.0121 | 9 | 58 | 13 | 1.0 | 0.2241 |
| AMAZON SO | 604 | 16 | 1.0 | 0.0265 | 10 | 1303 | 69 | 1.0 | 0.0530 |
| AMAZON TMI | 520 | 17 | 1.0 | 0.0327 | 11 | 138 | 40 | 1.0 | 0.2899 |
| TWITCH DE | 5742 | 382 | 1.0 | 0.0665 | 12 | 57 | 11 | 1.0 | 0.1930 |
| TWITCH ENGB | 3888 | 34 | 1.0 | 0.0874 | 13 | 63 | 25 | 1.0 | 0.3968 |
| TWITCH ES | 1360 | 160 | 1.0 | 0.1176 | 14 | 570 | 59 | 1.0 | 0.1035 |
| TWITCH FR | 2415 | 287 | 1.0 | 0.1188 | 15 | 257 | 52 | 1.0 | 0.2023 |
| TWITCH PTBR | 661 | 107 | 1.0 | 0.1619 | 16 | 254 | 24 | 1.0 | 0.0945 |
| TWITCH RU | 1075 | 85 | 1.0 | 0.0791 | 17 | 1572 | 79 | 1.0 | 0.0503 |
| DEEZER | 12538 | 32 | 1.0 | 0.0026 | 100 | 387 | 51 | 1.0 | 0.1318 |
| GITHUB | 9739 | 186 | 1.0 | 0.0191 | | | | | |
| TWITTER | 11355 | 156 | 1.0 | 0.0137 | | | | | |

In the experiments presented in Table 7, we set the target fairness level $\alpha = \frac{|\mathcal{S}\cap\mathcal{S}_p|}{|\mathcal{S}|} = 1$ in FADSG-I to extract the densest protected subset, and examined what fraction of the entire protected set it constitutes; i.e., if $\mathcal{S}$ is the densest protected subset obtained using FADSG-I, we evaluate the value of $\frac{|\mathcal{S}\cap\mathcal{S}_p|}{|\mathcal{S}_p|}$. As we can see from these tables, there are many cases where the solution comprises a small proportion of protected vertices (it can be even less than 5% - see GITHUB). Note that this stems from the structure of the protected set, since the solution that we get is always the densest subset of the protected set. Hence, FADSG-I may not be able to extract a dense subgraph that contains even 50% of the protected set - motivating the need, in some cases, for FADSG-II.

