# OpenReview forum: "Fairness-Aware Dense Subgraph Discovery"
_TMLR — Accepted by TMLR_

### Review · Reviewer_hAip · 2025-01-25

**Summary Of Contributions:**

The paper focuses on the problem of fair densest subgraph discovery and addresses the high computational complexity of previous methods by proposing two regularization terms and proving their computability. It introduces the POF metric to measure the trade-off for incorporating fairness. The optimization problems corresponding to the two regularization terms are formulated as the Densest Supermodular Subset (DSS) problem, which is approximately solved using the Super-Greedy++ algorithm, significantly reducing time complexity compared to solving via the maximum flow algorithm. Experiments on real-world datasets demonstrate the effectiveness of the proposed regularization terms in achieving fairness.

**Audience:**

Yes

**Broader Impact Concerns:**

The proposed method carries the risk of misuse in sensitive applications such as social network analysis, hiring networks, or political influence mapping. For instance, it could be exploited to justify biased decisions under the pretense of fairness by selectively defining fairness criteria or subgroup representations to suit specific agendas.

While the paper introduces multiple notions of fairness, these concepts are inherently context-dependent and may not universally align with ethical or widely accepted definitions of fairness. If the fairness criteria are poorly chosen or intentionally manipulated, the resulting dense subgraph may inadvertently reinforce existing biases or inequalities instead of addressing them.

**Claims And Evidence:**

Yes

**Requested Changes:**

1. Clarify the Scenarios for FADSG-I and FADSG-II:
   Provide a detailed explanation of the scenarios where FADSG-I fails to adequately represent protected nodes. Include examples or case studies from real-world datasets to illustrate these scenarios and discuss their prevalence.

2. Extend Fairness to Multi-Valued Sensitive Attributes:
   Discuss whether the proposed methods can be extended to handle multi-valued sensitive attributes. If possible, include a brief theoretical or experimental analysis to demonstrate the feasibility of such an extension.

3. Experimental Comparisons for FADSG-II:
   Include direct comparisons of FADSG-II with FADSG-I and baseline methods using the same metrics (e.g., POF and the proportion of protected nodes). If FADSG-II uses a different fairness criterion, explain why this criterion is necessary and how it relates to the metrics used for FADSG-I.

4. Highlight the Advantages of FADSG-I:
   Clearly articulate the advantages of FADSG-II over FADSG-I in terms of fairness, performance, or other criteria. Provide experimental evidence to support these claims.

**Strengths And Weaknesses:**

Strengths
1.Both FADSG-I and FADSG-II can be computed in polynomial time, overcoming the NP-hardness drawback of previous methods in the worst-case scenario.
2.FADSG-I and FADSG-II exhibit monotonicity, enabling the use of binary search to determine the strength of regularization.
3.The experimental results seem to validate the effectiveness of the two regularization methods.

Weaknesses
1.From the experimental results, FADSG-I appears capable of achieving a similar proportion of protected nodes being selected. What, then, is the advantage of FADSG-II over FADSG-I? The paper mentions that in certain scenarios, FADSG-I cannot adequately represent protected nodes. What are these scenarios, and how frequently do they occur in real-world applications?
2.The paper focuses on relatively simple fairness scenarios, considering only binary sensitive attributes. However, in real-world scenarios, sensitive attributes may involve multiple groups. Can the proposed methods be easily extended to handle multi-valued sensitive attributes?
3.In the experiments, why is FADSG-II not compared against FADSG-I and existing baseline methods? Although the paper mentions that FADSG-II uses a different fairness criterion, the metrics presented in the figures (POF and the proportion of protected nodes) are the same as those used for FADSG-I.

---

> ### Author Response · Authors · 2025-02-24
> **Response to reviewer hAip (1/3)**
>
> We thank the reviewer for their valuable time and effort in reviewing this paper. The comments are carefully addressed below.
>
> ### Weaknesses:
>
> > **Comment 1:** *From the experimental results, FADSG-I appears capable of achieving a similar proportion of protected nodes being selected. What, then, is the advantage of FADSG-II over FADSG-I? The paper mentions that in certain scenarios, FADSG-I cannot adequately represent protected nodes. What are these scenarios, and how frequently do they occur in real-world applications?*
>
> **Response:**\
> Please note that FADSG-I and FADSG-II promote a different notion of inclusion of protected vertices in the solution.
> A possible cause of confusion was that we set the target fairness value to be $\alpha=1/2$ for FADSG-I and $\delta=1$ for FADSG-II. When we set $\alpha=1/2$, we get $r_1(\mathcal{S})=1/2 \Leftrightarrow \frac{|\mathcal{S}\cap \mathcal{S}_p|}{|\mathcal{S}|} = \frac{1}{2}$. On the other hand, when we set $\delta=1$, we get $r_2(\mathcal{S})=1 \Leftrightarrow \frac{|\mathcal{S}\cap \mathcal{S}_p|}{|\mathcal{S}_p|} = \frac{1}{2}$ (Lemma 4). Notice the difference in the denominators, which implies that $r_1(\mathcal{S})$ corresponds to $50$\% of the solution being composed of protected nodes, whereas $r_2(\mathcal{S})$ corresponds to $50$\% of the total protected set being included in the solution. Clearly, the solutions need not be not the same.
>
> As explained in Section 3, our main motivation for considering FADSG-II is that it is a more focused formulation in terms of vertex inclusion, since for sufficiently large values of $\lambda$, the solution coincides with the *entire protected subset* $\mathcal{S}_p$. This means that we can always find a solution that contains all of the protected vertices using FADSG-II, which might not always be achievable with FADSG-I. This follows from the fact that for sufficiently large values of $\lambda$, the maximal level of representation attainable using FADSG-I corresponds to the *densest subset* of $\mathcal{S}_p$.
> This does not imply that FADSG-I *fails* to represent the protected vertices, but it ''may not provide satisfactory levels of representation'', depending on the needs of the application where FADSG is being used.
> Hence, FADSG-II is an additional tool for cases where we need more focused representation of the protected vertices in the solution.\
> In order to make the above point more clear,
> we have added a new section in Appendix H.3, which provides new experimental results for different datasets (Table 7). In these experiments, we set the target fairness level $\alpha = \frac{|\mathcal{S}\cap\mathcal{S}_p|}{|\mathcal{S}|}=1$ in FADSG-I to extract the densest protected subset, and examined what fraction of the entire protected set it constitutes; i.e., if $\mathcal{S}$ is the densest protected subset obtained using FADSG-I, we evaluate the value of $\frac{|\mathcal{S}\cap\mathcal{S}_p|}{|\mathcal{S}_p|}$.
> As we can see from these tables, there are many cases where the solution comprises a small proportion of protected vertices (it can be even less than $5$\%). Note that this stems from *the structure of the graph*, since the solution that we get is always the densest subset of the protected set. Hence, FADSG-I may not be able to extract a dense subgraph that contains even $50$\% of the protected set - motivating the need, in some cases, of FADSG-II.
>
> > **Comment 2:** *The paper focuses on relatively simple fairness scenarios, considering only binary sensitive attributes. However, in real-world scenarios, sensitive attributes may involve multiple groups. Can the proposed methods be easily extended to handle multi-valued sensitive attributes?*
>
> **Response:**\
> Please refer to our response to comment 1 of Reviewer SQso.

---

> > ### Author Response · Authors · 2025-02-24
> > **Response to reviewer hAip (2/3)**
> >
> > > **Comment 3:** *In the experiments, why is FADSG-II not compared against FADSG-I and existing baseline methods? Although the paper mentions that FADSG-II uses a different fairness criterion, the metrics presented in the figures (POF and the proportion of protected nodes) are the same as those used for FADSG-I.*
> >
> > **Response:**\
> > As clarified previously, FADSG-I and FASDSG-II employ different fairness criteria. Based of the target fairness values in each case, each formulation guarantees an inclusion of vertices in the solution in a different manner. Hence, the ''proportion of protected nodes'' is different for each case - the denominators in the y-axis of the figures for FADSG-I (Figs 2, 4, 5, 6, 7), $\frac{|\mathcal{S}\cap \mathcal{S}_p|}{|\mathcal{S}|}$, and for FADSG-II (Fig 8), $\frac{|\mathcal{S}\cap \mathcal{S}_p|}{|\mathcal{S}_p|}$, as we explain in Section 3.
> > Regarding the evaluation criteria, the definition of PoF is indeed the same for both, but it is measured w.r.t. the target fairness level, which, as we mentioned, differs between the two formulations. Hence, an ''apples-to-apples'' comparison cannot be made between FADSG-I and FASDSG-II.
> >
> > Furthermore, the fairness criteria of the prior methods are defined w.r.t. $\frac{|\mathcal{S}\cap \mathcal{S}_p|}{|\mathcal{S}|}$ (same as FADSG-I), and are not directly comparable to FADSG-II. Hence, we cannot do an ''apples-to-apples'' comparison between FADSG-II and prior works either.
> >
> > ### Requested Changes:
> >
> > > **Comment 5:** *Clarify the Scenarios for FADSG-I and FADSG-II:\
> > Provide a detailed explanation of the scenarios where FADSG-I fails to adequately represent protected nodes. Include examples or case studies from real-world datasets to illustrate these scenarios and discuss their prevalence.*
> >
> > **Response:**\
> > Please see our response to your first comment.
> >
> > > **Comment 6:** *Extend Fairness to Multi-Valued Sensitive Attributes: Discuss whether the proposed methods can be extended to handle multi-valued sensitive attributes. If possible, include a brief theoretical or experimental analysis to demonstrate the feasibility of such an extension.*
> >
> > **Response:**\
> >  Please see our response to the first comment of Reviewer SQso. Since we are presently working on the extension to the multi-group setting using very different algorithmic techniques, we have mentioned in Conclusions that the multi-group setting constitutes a direction of future work.
> >
> > > **Comment 7:** *Experimental Comparisons for FADSG-II:\
> > Include direct comparisons of FADSG-II with FADSG-I and baseline methods using the same metrics (e.g., POF and the proportion of protected nodes). If FADSG-II uses a different fairness criterion, explain why this criterion is necessary and how it relates to the metrics used for FADSG-I.*
> >
> > **Response:**\
> > Please refer to our response to your first comment where we have explained the difference between the two fairness criteria, and why FADSG-II may be needed in some cases. Additionally, Lemma 5 (previously 2) specifies a relationship between the regularizers of FADSG-I and FADSG-II.
> > Finally, our response to your third comment explains why a direct comparison cannot be made between FADSG-I and FADSG-II. We have included some of these responses in the manuscript.
> >
> > > **Comment 8:** *Highlight the Advantages of FADSG-I:\
> > Clearly articulate the advantages of FADSG-II over FADSG-I in terms of fairness, performance, or other criteria. Provide experimental evidence to support these claims.*
> >
> > **Response:**\
> > Please refer to our response to your 1st and 5th comments.

---

> > > ### Author Response · Authors · 2025-02-24
> > > **Response to reviewer hAip (3/3)**
> > >
> > > ### Broader Impact Concerns:
> > >
> > > > **Comment:** *The proposed method carries the risk of misuse in sensitive applications such as social network analysis, hiring networks, or political influence mapping. For instance, it could be exploited to justify biased decisions under the pretense of fairness by selectively defining fairness criteria or subgroup representations to suit specific agendas.\
> > > While the paper introduces multiple notions of fairness, these concepts are inherently context-dependent and may not universally align with ethical or widely accepted definitions of fairness. If the fairness criteria are poorly chosen or intentionally manipulated, the resulting dense subgraph may inadvertently reinforce existing biases or inequalities instead of addressing them.*
> > >
> > > **Response:**\
> > >  We note that the group fairness formulations considered herein do align with pre-existing notions of fairness (please see our response to the last requested change of Reviewer RYxZ). In fact, they can be viewed as finer-grained notions compared to demographic parity or perfect balance, as they offer greater flexibility in attaining a target level of representation. \
> > > On the other hand, the reviewer is right in that the proposed methods require appropriate specification of the protected set. Thus, if a user exercises poor judgment/is unscrupulous in choosing a protected set based on an inappropriate attribute, then the solution could reinforce existing biases. However, as long as the choice of the protected set is publicly disclosed, then everyone can make an informed decision on whether to accept the results, and the reviewer's concern will be mitigated. We would also like to point out that similar comments apply to any method that can be used to protect different subgroups.

---

### Review · Reviewer_RYxZ · 2025-02-01

**Summary Of Contributions:**

This paper makes three significant contributions to algorithmic fairness in graph mining:
- First, it introduces two novel, polynomial-time solvable formulations for fair dense subgraph discovery (DSD), addressing scenarios where graph vertices contain sensitive attributes like gender or race. Unlike previous NP-hard approaches, these formulations are computationally tractable.
- Second, through carefully designed regularizers, the methods enable flexible control over fairness levels, allowing the balance between graph density and fair representation of protected groups.
- Third, the authors analyze the trade-off between subgraph density and fairness through their "price of fairness" metric, quantifying the density sacrifice needed for achieving desired fairness levels. They validate their approach through comprehensive empirical evaluation on multiple datasets.

**Audience:**

Yes

**Broader Impact Concerns:**

This work provided valuable insights on the responsible algorithms for dense graph detection. I do not foresee any potential ethical concerns.

**Claims And Evidence:**

Yes

**Requested Changes:**

- Please put all the problem formulations in a dedicated problem formulation section. It was a pain when I was confused about the definition of densest subgraph problem, but ended up finding it in the related work section. Also, please highlight the definitions of Equation (1) and Equation (3).
- I would suggest the authors to further clarify the connections between the two fairness-aware densest subgraph problems and the conventional fairness measures like demographic parity and equality of opportunity.

**Strengths And Weaknesses:**

**Strengths**
The paper's primary contribution lies in its novel approach to incorporating fairness considerations into dense subgraph discovery through regularization. The authors provided rigorous theoretical analysis in presenting their formulations and provide valuable insights regarding the effects of regularization parameters. In general, the manuscript is well-structured and systematic in its presentation. I also appreciate their comprehensive theoretical analysis of fairness-density trade-offs and provide the price of fairness bounds.

**Weakness**
I have concerns regarding the theoretical soundness of the proposed formulations in Equations (1) and (3).  In Equation (1), the regularizer $r_1(x)$ is designed to encourage inclusion of vertices from the protected subset $S_p$. However, this formulation may have an unintended consequence: at any given density level, selecting vertices exclusively from $S_p$ could potentially yield the optimal solution, as there are no constraints preventing this outcome. Similarly, in Equation (3), minimizing the penalized term could be achieved simply by selecting vertices predominantly from $S_p$, potentially undermining the intended balance between protected and unprotected groups. How could these formulations still promote fairness that the numbers of vertices from protected set and unprotected set are balanced?

---

> ### Author Response · Authors · 2025-02-24
> **Response to reviewer RYxZ**
>
> We thank the reviewer for their valuable time and effort in reviewing this paper. The comments are carefully addressed below.
>
> ### Weaknesses:
>
> > **Comment:** *I have concerns regarding the theoretical soundness of the proposed formulations in Equations (1) and (3). In Equation (1), the regularizer $r_1(x)$  is designed to encourage inclusion of vertices from the protected subset $\mathcal{S}_p$. However, this formulation may have an unintended consequence: at any given density level, selecting vertices exclusively from $\mathcal{S}_p$ could potentially yield the optimal solution, as there are no constraints preventing this outcome. Similarly, in Equation (3), minimizing the penalized term could be achieved simply by selecting vertices predominantly from $\mathcal{S}_p$, potentially undermining the intended balance between protected and unprotected groups. How could these formulations still promote fairness that the numbers of vertices from protected set and unprotected set are balanced?*
>
> **Response:**\
> To clarify, in our formulations, we specify a desired *fairness level* that the solution must satisfy, rather than a *density level*. Specifically, in the formulation of equation (2) (previously (1)), FADSG-I, the fairness level is explicitly defined as the proportion of vertices in the resulting subgraph that belong to the protected subset $\mathcal{S}_p$, ensuring that fairness is directly controlled rather than emerging as a side effect. A solution consisting exclusively of vertices from $\mathcal{S}_p$ can only occur when the target fairness level is explicitly set to $\alpha = 1$, which enforces this outcome. Thus, the formulation does not inherently favor selecting only from $\mathcal{S}_p$ unless that is the specified objective.\
> For the formulation in equation (4) (previously (3)), i.e., FADSG-II, if we set $\delta=1$, and we solve the problem, this will result in $r_2(\mathcal{S}) = 1$, which is equivalent to $\frac{|\mathcal{S}\cap \mathcal{S}_p|}{|\mathcal{S}_p|} = \frac{1}{2}$, i.e., $50$\% of the protected vertices will be included in the resulting subgraph $\mathcal{S}$. Similarly to the previous formulation, we *pre-specify* the target fairness level, thus we get a solution satisfying this fairness level.
> Hence, the formulations can promote balanced solutions by appropriately tuning the target fairness levels $\alpha$, for FADSG-I, and $\delta$, for FADSG-II. For a more detailed explanation about the selection of the target fairness levels please refer to Section 3.

---

> > ### Author Response · Authors · 2025-02-24
> > **Response to reviewer RYxZ - continued**
> >
> > ### Requested Changes:
> >
> > > Comment: *Please put all the problem formulations in a dedicated problem formulation section. It was a pain when I was confused about the definition of densest subgraph problem, but ended up finding it in the related work section. Also, please highlight the definitions of Equation (1) and Equation (3).*
> >
> > **Response:**\
> > We apologize for the confusion. We included the formulation of the classic DSG problem at the beginning of Section 3.\
> > Moreover, we highlighted the definitions of Equations.\
> > Please see the updated Section 3.
> >
> > > **Comment:** *I would suggest the authors to further clarify the connections between the two fairness-aware densest subgraph problems and the conventional fairness measures like demographic parity and equality of opportunity.*
> >
> > **Response:**\
> > FADSG-I and FADSG-II promote a different notion of inclusion of protected vertices to the solution.\
> > In order to make the distinction more clear we will present some specific cases for the two formulations. A possibly confusing case can be the one that we set for FADSG-I the target fairness value to be $\alpha=1/2$ and for FADSG-II $\delta=1$. When we set $\alpha=1/2$, we get $r_1(\mathcal{S})=1/2 \Leftrightarrow \frac{|\mathcal{S}\cap \mathcal{S}_p|}{|\mathcal{S}|} = \frac{1}{2}$. When we set $\delta=1$, we get (Lemma 4) $r_2(\mathcal{S})=1 \Leftrightarrow \frac{|\mathcal{S}\cap \mathcal{S}_p|}{|\mathcal{S}_p|} = \frac{1}{2}$. Notice the difference in the denominators.\
> > Furthermore, the connection between the two regularizers of FADSG-I and II can be seen in Figure 9, in Appendix A.2.
> >
> > Regarding demographic parity, we can achieve it if we set the target fairness level of FADSG-I to be equal to the portion of protected vertices in the whole graph, i.e., $\alpha=n_p/n$. We also included it in the text after the definition of FADSG-I. \
> > On the other hand, equality of odds and equality of opportunity are fairness measures more appropriate in classification scenarios in contrast to our unsupervised setting.
> > This becomes evident if we take a look at the definitions of Dong et al., 2023, in binary node classification. Given a  predicted label $\hat{Y}$, the sensitive attribute $S$, and the ground truth label $Y$, the demographic parity criterion is formulated as
> > $$
> >     P(\hat{Y}=1|S=0) = P(\hat{Y}=1|S=1),
> > $$
> > the equality of odds as
> > $$
> >     P(\hat{Y}=1|S=0,Y=y) = P(\hat{Y}=1|S=1,Y=y),
> > $$
> > and the equality of opportunity (equality of odds only for positive predictions)
> > $$
> >     P(\hat{Y}=1|S=0,Y=1) = P(\hat{Y}=1|S=1,Y=1).
> > $$
> > It is evident that if the sensitive attribute coincides with the ground truth label - as it is in our case - then equality of odds and equality of opportunity coincide with demographic parity.

---

### Review · Reviewer_SQso · 2025-02-04

**Summary Of Contributions:**

The paper is concerned with the densest subgraph (DSD) problem. Unlike prior work it introduces two objectives that tradeoff the value of DSD objective and fairness using a regularization parameter. Further, unlike prior work the optimization is tractable. Finally, the paper shows better performance over a collection of baselines experimentally.

**Audience:**

Yes

**Broader Impact Concerns:**

I don't think that there are broader impact concerns.

**Claims And Evidence:**

Yes

**Requested Changes:**

1-In Section 5 for PoF, I don’t see the point behind lollipop example? How is it that useful since it is a very specific example? Also, a figure would improve presentation here.


2-Please add an explanation on how is this setting significantly different from for example Miyauchi? Specifically, what if the algorithm from Miyauchi is run at different fraction values using binary search?

3-As mentioned under weaknesses explain the tractability advantage of your setting given that you are using approximation algorithms

**Strengths And Weaknesses:**

## Strengths:

-The paper generalizes the prior work and introduces two meaningful problems that have a compromise between DSD and fairness using a parameter  $\lambda$


-The paper includes experiments and comparison to baselines which show an advantage.


## Weaknesses:

-The paper is only using two groups which is a limitation.

-It seems that the paper claims that it straightforwardly follows that the problem is solvable in polynomial time for any $\lambda$ using max flow but where is this proof/explanation?

-From Section 6, the problem is solved using binary search and a fast approximate algorithm. But this makes the paper loose the claimed advantage over prior work since they involve an NP-hard problem that cannot be solved exactly.

-It would be better if there was also an upper bound on $r_2$ using $r_1$.

-The mathematical derivations do not seem very informative/substantial,e.g., Lemma 1, Proposition 3.

Minor point: using $\boldsymbol{1}$ as a vector of ones is more clear than using $\boldsymbol{e}$.

---

> ### Author Response · Authors · 2025-02-24
> **Response to reviewer SQso (1/4)**
>
> We thank the reviewer for their valuable time and effort in reviewing this paper. Please find our response below.
>
> ### Weaknesses
>
> > **Comment 1:** *The paper is only using two groups which is a limitation.*
>
> **Response:**\
> While we indeed consider only two groups, our approach is the first to tackle this foundational setting in a principled manner, serving as a stepping-stone for analyzing the more challenging scenario involving multiple protected groups. A natural extension of FADSG-I and II to the case of $K \geq 2 $ protected groups would be to add multiple regularization terms, one per protected group $k \in [K]:= \{1,\cdots,K\}$. For example, the multi-group extension of FADSG-I would be
> $$
> \max_ {\mathbf{x} \in \{0,1\}^n} ~ \biggl\\{
>     g(\mathbf{x},\lambda): = \frac{\mathbf{x}^T\mathbf{Ax}}{\mathbf{e}^T\mathbf{x}} + \sum_ {k=1}^K \lambda_ k \cdot \frac{\mathbf{e}_ {\mathcal{S}_ k}^T\mathbf{x}}{\mathbf{e}^T\mathbf{x}}
> \biggr\\},
> $$
> while that corresponding to FADSG-II
> $$
> \max_ {\mathbf{x} \in \{0,1\}^n} ~ \biggl\\{
>     h(\mathbf{x},\lambda): = \frac{\mathbf{x}^T\mathbf{Ax}}{\mathbf{e}^T\mathbf{x}} - \sum_ {k=1}^K \lambda_ k \cdot \frac{\||\mathbf{e}_ {\mathcal{S}_ k}-\mathbf{x}\||^2}{\mathbf{e}^T\mathbf{x}}
> \biggr\\}.
> $$
> In the above formulations, we have $K$ non-overlapping groups $\\{\mathcal{S}_ k\\}_ {k=1}^K$, and
> $\mathbf{e}_ {\mathcal{S}_ k}$ denotes the indicator vector of protected group $k \in [K]$. Additionally, $\\{\lambda_ k\\}_ {k=1}^K$ denote the corresponding regularization parameters.
> Since these problems still correspond to maximizing the ratio of a supermodular function and a modular function, a straightforward extension of the arguments in Appendix B guarantees tractability.
> However, tuning the regularization parameters $\\{\lambda_k\\}_{k=1}^K$ to attain a target fairness level is significantly more challenging due to the increased degrees of freedom.
> Moreover, these ''sum''-fairness criteria cannot guarantee a sufficient representation for each one of the protected groups - for example, simply ensuring that the sum of the regularizers is large in the first equation does not imply that each group $\mathcal{S}_k$ is well represented in the solution.
> To address these issues, we are currently working on extending our approach to handle multiple vertex groups using regularizers which are designed to ensure that no protected group is inadequately represented - utilizing a notion of
> min-fairness motivated by the Rawlsian principle of social justice [1].
> However, the resulting problems are much harder (i.e., not tractable in the worst-case), and consequently require very different algorithmic strategies for approximation compared to the techniques adopted herein. Fleshing out these challenges and solutions is naturally beyond the scope of this paper, and merits exploration in separate follow-up work.
>
> [1] J. Rawls. A theory of justice. Applied ethics. Routledge, 2017.
>
>
> > **Comment 2:**  *It seems that the paper claims that it straightforwardly follows that the problem is solvable in polynomial time for any $\lambda$ using max flow but where is this proof/explanation?*
>
> **Response:**\
> Indeed, we did not provide any proof of optimality using max-flows, since we found it quite straightforward following known results in this area [Goldberg 1984, Lanciano et al. 2024].
> Following your suggestion, we have included additional notes in Appendix B.1, showing how max-flows can be applied to optimally solve FADSG-I and FADSG-II (for a fixed $\lambda$), respectively.

---

> ### Author Response · Authors · 2025-02-24
> **Response to reviewer SQso (2/4)**
>
> > **Comment 3:** *From Section 6, the problem is solved using binary search and a fast approximate algorithm. But this makes the paper loose the claimed advantage over prior work since they involve an NP-hard problem that cannot be solved exactly.*
>
> **Response:**\
> Although we indeed employ approximation, please note this is not the same as approximating the NP--hard problems of the prior art. More specifically, we employ approximation to improve scalability, as opposed to dealing with intractability.
> For a fixed $\lambda$, both of our formulations can be solved exactly using max flows, albeit at high complexity (see Section 6). In order to address this, we utilize a fast approximation algorithm (Super-Greedy++), which provides a $(1-\epsilon)$-approximation on the optimal value of FADSG-I in $O((m+n\log n)/\epsilon^2)$ time. Hence, for a fixed value of $\lambda$, we can solve FADSG-I to *any* level of desired accuracy in polynomial-time.
> In this way, by varying $\lambda$, we can closely trace the Pareto-optimal frontier between density and our chosen fairness criterion.
> However, our end objective is to select a point on this frontier; i.e., find the densest subgraph which attains a desired target fairness level, for which we use bisection on $\lambda$.
> Since we use Super-Greedy++ with bisection, the final result need not be the same as that obtained by solving each instance of FADSG-I using max-flows. To test the difference in performance between bisection-search using max-flows and Super-Greedy++, we performed an empirical comparison using the implementation of flow-based DSG available at https://pypi.org/project/dsd, and adapted it for FADSG-I. We ran experiments for some small networks for which it is feasible to run max-flow (PolBlogs, TwitchPTBR-ES), and in all cases we get very similar values that give subgraphs with nearly identical densities and representation. You can find the associated results in the following table.
>
> | **Comparison of Max-Flow solution $\mathcal{S}_ {\text{flow}}$ and Super-Greedy++ solution $\mathcal{S}_ {\text{++}}$ for FADSG-I.** | | | | | | |
> |---|---|---|---|---|---|---|
> | **PolBlogs** | | | | | | |
> | $\alpha$ | 0.4 | 0.5 | 0.6 | 0.7 | 0.8 | 0.9 |
> | $ \|\mathcal{S}_ {\text{flow}}\|$ | 283  | 283  | 280  | 280  | 280  | 159  |
> | $ \|\mathcal{S}_ {\text{++}}\|$   | 283  | 277  | 287  | 287  | 287  | 159  |
> | $ \|\mathcal{S}_ {\text{flow}} \cap \mathcal{S}_ {\text{++}}\|$ | 283  | 277  | 279  | 279  | 279  | 159  |
> | $ \|\mathcal{S}_ {\text{flow}} \cap \mathcal{S}_ {p}\| / \|\mathcal{S}_ {\text{flow}}\|$ | 0.399 | 0.495 | 0.564 | 0.564 | 0.564 | 0.906 |
> | $ \|\mathcal{S}_ {\text{++}} \cap \mathcal{S}_ {p}\| / \|\mathcal{S}_ {\text{++}}\|$ | 0.399 | 0.491 | 0.578 | 0.578 | 0.578 | 0.906 |
> | **Twitch ptbr** | | | | | | |
> | $\alpha$ | 0.4 | 0.5 | 0.6 | 0.7 | 0.8 | 0.9 |
> | $ \|\mathcal{S}_ {\text{flow}}\|$  | —  | 337  | 296  | 291  | 287  | 233  |
> | $ \|\mathcal{S}_ {\text{++}}\|$   | —  | 333  | 294  | 287  | 286  | 233  |
> | $ \|\mathcal{S}_ {\text{flow}} \cap \mathcal{S}_ {\text{++}}\|$ | —  | 333  | 294  | 287  | 282  | 229  |
> | $ \|\mathcal{S}_ {\text{flow}} \cap \mathcal{S}_ {p}\| / \|\mathcal{S}_ {\text{flow}}\|$ | —  | 0.496 | 0.608 | 0.694 | 0.794 | 0.893 |
> | $ \|\mathcal{S}_ {\text{++}} \cap \mathcal{S}_ {p}\| / \|\mathcal{S}_ {\text{++}}\|$ | —  | 0.502 | 0.609 | 0.704 | 0.808 | 0.901 |
> | **Twitch es** | | | | | | |
> | $ \alpha $ | 0.4 | 0.5 | 0.6 | 0.7 | 0.8 | 0.9 |
> | $ \|\mathcal{S}_ {\text{flow}}\|$ | 554  | 579  | 581  | 574  | 527  | 434  |
> | $ \|\mathcal{S}_ {\text{++}}\|$   | 553  | 577  | 556  | 568  | 495  | 433  |
> | $ \|\mathcal{S}_ {\text{flow}} \cap \mathcal{S}_ {\text{++}}\|$ | 553  | 571  | 556  | 562  | 495  | 428  |
> | $ \|\mathcal{S}_ {\text{flow}} \cap \mathcal{S}_ {p}\| / \|\mathcal{S}_ {\text{flow}}\|$ | 0.394 | 0.496 | 0.606 | 0.702 | 0.803 | 0.896 |
> | $ \|\mathcal{S}_ {\text{++}} \cap \mathcal{S}_ {p}\| / \|\mathcal{S}_ {\text{++}}\|$ | 0.394 | 0.504 | 0.606 | 0.688 | 0.800 | 0.891 |
>
> Meanwhile, for FADSG-II, even though we cannot theoretically guarantee that Super-Greedy++ does as good a job in approximating the max-flow solution for a fixed $\lambda$, this does not detract from the fact that the problem is polynomial-time.
>
> A more subtle issue is that for the bisection framework to provably terminate, the functions $\psi_1(\lambda)$ and $\psi_2(\lambda)$ (defined in Section 4.1) should be continuous for every target fairness level, and for every graph. Although we cannot show this at present, we observed in our experiments that depending on the nature of the underlying Pareto-optimal frontier, some fairness levels may be more difficult to attain than others. However, in general, the procedure works very well on real datasets.

---

> ### Author Response · Authors · 2025-02-24
> **Response to reviewer SQso (3/4)**
>
> > **Comment 4:** *It would be better if there was also an upper bound on $r_2$ using $r_1$.*
>
> **Response:**\
> Thank you for the suggestion. We have derived an upper bound as well: $r_2(S) \leq 1+ \frac{n_p}{2} - 2r_1(S)$. We have updated Lemma 5 in the main text and we have included a clearer interpretation of the relationship between $r_1(\cdot)$ and $r_2(\cdot)$; see Figure 9 in Appendix A.2.
>
>
> > **Comment 5:** *The mathematical derivations do not seem very informative/substantial, e.g., Lemma 1, Proposition 3.*
>
> **Response:**\
> Lemma 4 (previously Lemma 1) spells out what value the regularizer $r_2$ should take ($r_2(S)=1$) in order for the solution of FADSG-II to satisfy the target fairness requirement of having $50$\% of the protected vertices to be included in the solution. In that regard, it is an important result. Indeed, this is the target fairness level that we set in our experiments with FADSG-II.
>
> Meanwhile, Proposition 6 (previously 3) is an argument which shows that there exists a
> finite value of $\lambda_{\max}$ for FADSG-I, such that for any $\lambda\geq\lambda_{\max}$, the solution will be the densest protected subset of $S_p$. We know that such a $\lambda$ should exist, since the function $f(\lambda) = \max_{\mathbf{x} \in \\{0,1\\}^n}  g(\mathbf{x},\lambda)$ is piece-wise affine, and hence has a finite number of break-points, the last of which corresponds to $\lambda_{\max}$ (i.e., the end of the regularization path). In this regard, Proposition 6 provides a lower bound on how large $\lambda_{\max}$ can be.
> However, it is difficult to evaluate the lower bound exactly. Hence, in practice, we resorted to trial and error to find a sufficiently large $\lambda$ for bisection-search, and we present the values that we used in our experiments in Table 6 at Appendix G.
>
> > **Comment 6:** *Minor point: using $\mathbf{1}$ as a vector of ones is more clear than using $\mathbf{e}$.*
>
> **Response:**\
> Thank you for your suggestion. Since we are also using $\mathbf{e}_i$ to denote the indicator vector of vertex $i$, we feel that using $\mathbf{e}$ as the vector of all ones would be more consistent with our present notation. Hence, would prefer to leave it as is.

---

> ### Author Response · Authors · 2025-02-24
> **Response to reviewer SQso (4/4)**
>
> ### Requested Changes
>
> > **Comment 7:** *In Section 5 for PoF, I don't see the point behind lollipop example? How is it that useful since it is a very specific example? Also, a figure would improve presentation here.*
>
> **Response:**\
> We present the lollipop example to shed light on the fundamental trade-off between fairness and subgraph density. Since these two quantities can be in tension with each other, it is quite natural to consider how improvements in fairness come at the expense of loss in subgraph density. We note that none of the prior art in fair dense subgraph discovery quantified this trade-off, which is why we considered this problem in the first place.
> As we mention in Section 5, the lollipop graph is not completely contrived - in fact, it mimics two properties of real-world graphs in that (i): it is sparse, and (ii): the majority of vertices have low degree, while the remaining ones have high degree. Through this example, we showcase that when we include fairness into the DSG problem, high PoF values can naturally occur depending on the structure of the graph.
>
> Note that in Section 5 we included Figure 3 with a toy example of lollipop for $n=16$, to facilitate visualization.
>
> > **Comment 8:** *Please add an explanation on how is this setting significantly different from for example Miyauchi? Specifically, what if the algorithm from Miyauchi is run at different fraction values using binary search?*
>
> **Response:**\
> The key difference between our setting and that of Miyauchi et al. lies in how we promote fairness. Our approach utilizes optimization with fairness promoting regularizers (Dong et al. 2023, Section 4.1), whereas that of Miyauchi et al. enforces fairness via hard constraints (Dong et al. 2023, Section 4.2). In our setting, we employ binary search on the regularization parameter $\lambda$ in order to extract the densest subgraph that attains a predefined target fairness level. In contrast, binary search is not applicable for the framework of Miyauchi et al., since they do not have any regularization parameter to tune.
>
> Furthermore, in our setting we set an *exact* target fairness level that we want to achieve. For example, for FADSG-I, when we set $\alpha=0.7 \rightarrow r_1(S)=0.7 \Leftrightarrow \frac{|S\cap S_p|}{|S|} = 0.7$, we are trying to find the densest subgraph such that $70$\% of its vertices are protected. On the other hand, the formulation of Miyauchi et al. only guarantees that the representation level of each subgroup is *no more* than a fraction of the total vertices in the solution; i.e., given $K$ different vertex groups, Miyauchi et al. aim to find the densest subgraph that obeys the following fairness constraints
> $$
>     \frac{|S \cap S_k|}{|S|} \leq \alpha, \forall ~k \in [K].
> $$
> It is evident that the above constraints do not guarantee *a priori* which subgroup $S_k$ attains the maximum or whether this upper bound is attained at all. This implies that even in the simplest setting of two subgroups (i.e., protected and unprotected), the method may not be effective in guaranteeing a target level of representation of the protected group, except of the case for $\alpha=0.5$ which coincides with the *perfect balance* setting studied in Anagnostopoulos et al.
>
> > **Comment 9:** *As mentioned under weaknesses explain the tractability advantage of your setting given that you are using approximation algorithms.*
>
> **Response:**\
> Thank you for your suggestion. We have included the last paragraph in Section 2, based on our response to Comment 3.

---

### Author Response · Authors · 2025-02-24
**General Response**

We would like to thank all of the reviewers for their valuable effort and suggestions.\
We have updated our paper based on their comments and requests. The updated version is now available; with all changes are marked in blue. Additionally, we have also increased the size of the figures to improve visibility.

---

### Decision · Action_Editor_gCeH · 2025-03-21

**Recommendation:** Accept as is

**Comment:**

The current version of this paper is self-standing, with clear motivations, theoretical analyses, and experimental verification. With the excellent review team, I mainly agree with their comments but have a minor issues on the figure visualization. I suggest the authors improving the figures in terms of figure size (some figures are too small) and removing the necessary background color.

**Audience:**

Fairness is an increasing topic in the machine learning community.

**Claims And Evidence:**

This is a self-standing paper considering the fairness-aware dense subgraph discovery. Here the self-standing means the claims made in this paper is well supported by convincing and clear evidence.

I have a great review team, which consists of three experts in the fairness-focused machine learning area and provides high-quality comments. The authors also address these comments well.